# Gaussian Process Bandit Optimisation with Multi-fidelity Evaluations

**Kirthevasan Kandasamy** [♮], **Gautam Dasarathy** [◇], **Junier Oliva** [♮],
**Jeff Schneider** [♮], **Barnabás Póczos** [♮]
[♮] Carnegie Mellon University, [◇] Rice University
{kandasamy, joliva, schneide, bapoczos}@cs.cmu.edu, gautamd@rice.edu

## Abstract

In many scientific and engineering applications, we are tasked with the optimisation of an expensive to evaluate black box function $f$. Traditional methods for this problem assume just the availability of this single function. However, in many cases, cheap approximations to $f$ may be obtainable. For example, the expensive real world behaviour of a robot can be approximated by a cheap computer simulation. We can use these approximations to eliminate low function value regions cheaply and use the expensive evaluations of $f$ in a small but promising region and speedily identify the optimum. We formalise this task as a *multi-fidelity* bandit problem where the target function and its approximations are sampled from a Gaussian process. We develop MF-GP-UCB, a novel method based on upper confidence bound techniques. In our theoretical analysis we demonstrate that it exhibits precisely the above behaviour, and achieves better regret than strategies which ignore multi-fidelity information. MF-GP-UCB outperforms such naive strategies and other multi-fidelity methods on several synthetic and real experiments.

## 1   Introduction

In stochastic bandit optimisation, we wish to optimise a *payoff* function $f : \mathcal{X} \to \mathbb{R}$ by sequentially querying it and obtaining *bandit feedback*, i.e. when we query at any $x \in \mathcal{X}$, we observe a possibly noisy evaluation of $f(x)$. $f$ is typically expensive and the goal is to identify its maximum while keeping the number of queries as low as possible. Some applications are hyper-parameter tuning in expensive machine learning algorithms, optimal policy search in complex systems, and scientific experiments [20, 23, 27]. Historically, bandit problems were studied in settings where the goal is to maximise the cumulative reward of all queries to the payoff instead of just finding the maximum. Applications in this setting include clinical trials and online advertising.

Conventional methods in these settings assume access to only this single expensive function of interest $f$. We will collectively refer to them as *single fidelity* methods. In many practical problems however, cheap approximations to $f$ might be available. For instance, when tuning hyper-parameters of learning algorithms, the goal is to maximise a cross validation (CV) score on a training set, which can be expensive if the training set is large. However CV curves tend to vary smoothly with training set size; therefore, we can train and cross validate on small subsets to approximate the CV accuracies of the entire dataset. For a concrete example, consider kernel density estimation (KDE), where we need to tune the bandwidth $h$ of a kernel. Figure 1 shows the CV likelihood against $h$ for a dataset of size $n = 3000$ and a smaller subset of size $n = 300$. The two maximisers are different, which is to be expected since optimal hyper-parameters are functions of the training set size. That said, the curve for $n = 300$ approximates the $n = 3000$ curve quite well. Since training/CV on small $n$ is cheap, we can use it to eliminate bad values of the hyper-parameters and reserve the expensive experiments with the entire dataset for the promising candidates (e.g. boxed region in Fig. 1).

In online advertising, the goal is to maximise the cumulative number of clicks over a given period. In the conventional bandit treatment, each query to $f$ is the display of an ad for a specific time, say one

hour. However, we may display ads for shorter intervals, say a few minutes, to approximate its hourly performance. The estimate is biased, as displaying an ad for a longer interval changes user behaviour, but will nonetheless be useful in gauging its long run click through rate. In optimal policy search in robotics and automated driving vastly cheaper computer simulations are used to approximate the expensive real world performance of the system. Scientific experiments can be approximated to varying degrees using less expensive data collection, analysis, and computational techniques.

In this paper, we cast these tasks as *multi-fidelity bandit optimisation* problems assuming the availability of cheap approximate functions (fidelities) to the payoff $f$. **Our contributions** are:

1. We present a formalism for multi-fidelity bandit optimisation using Gaussian Process (GP) assumptions on $f$ and its approximations. We develop a novel algorithm, Multi-Fidelity Gaussian Process Upper Confidence Bound (MF-GP-UCB) for this setting.
2. Our theoretical analysis proves that MF-GP-UCB explores the space at lower fidelities and uses the high fidelities in successively smaller regions to zero in on the optimum. As lower fidelity queries are cheaper, MF-GP-UCB has better regret than single fidelity strategies.
3. Empirically, we demonstrate that MF-GP-UCB outperforms single fidelity methods on a series of synthetic examples, three hyper-parameter tuning tasks and one inference problem in Astrophysics. Our matlab implementation and experiments are available at github.com/kirthevasank/mf-gp-ucb.

**Related Work:** Since the seminal work by Robbins [25], the multi-armed bandit problem has been studied extensively in the $K$-armed setting. Recently, there has been a surge of interest in the optimism under uncertainty principle for $K$ armed bandits, typified by upper confidence bound (UCB) methods [2, 4]. UCB strategies have also been used in bandit tasks with linear [6] and GP [28] payoffs. There is a plethora of work on single fidelity methods for global optimisation both with noisy and noiseless evaluations. Some examples are branch and bound techniques such as dividing rectangles (DiRect) [12], simulated annealing, genetic algorithms and more [17, 18, 22]. A suite of single fidelity methods in the GP framework closely related to our work is Bayesian Optimisation (BO). While there are several techniques for BO [13, 21, 30], of particular interest to us is the Gaussian process upper confidence bound (GP-UCB) algorithm of Srinivas et al. [28].

Many applied domains of research such as aerodynamics, industrial design and hyper-parameter tuning have studied multi-fidelity methods [9, 11, 19, 29]; a plurality of them use BO techniques. However none of these treatments neither formalise nor analyse any notion of *regret* in the multi-fidelity setting. In contrast, MF-GP-UCB is an intuitive UCB idea with good theoretical properties. Some literature have analysed multi-fidelity methods in specific contexts such as hyper-parameter tuning, active learning and reinforcement learning [1, 5, 26, 33]. Their settings and assumptions are substantially different from ours. Critically, none of them are in the more difficult bandit setting where there is a price for exploration. Due to space constraints we discuss them in detail in Appendix A.3.

The multi-fidelity poses substantially new theoretical and algorithmic challenges. We build on GP-UCB and our recent work on multi-fidelity bandits in the $K$-armed setting [16]. Section 2 presents our formalism including a notion of regret for multi-fidelity GP bandits. Section 3 presents our algorithm. The theoretical analysis is in Appendix C with a synopsis for the 2-fidelity case in Section 4. Section 6 presents our experiments. Appendix A.1 tabulates the notation used in the manuscript.

## 2 Preliminaries

We wish to maximise a payoff function $f : \mathcal{X} \to \mathbb{R}$ where $\mathcal{X} \equiv [0, r]^d$. We can interact with $f$ only by querying at some $x \in \mathcal{X}$ and obtaining a noisy observation $y = f(x) + \epsilon$. Let $x_\star \in \operatorname{argmax}_{x \in \mathcal{X}} f(x)$ and $f_\star = f(x_\star)$. Let $\mathbf{x}_t \in \mathcal{X}$ be the point queried at time $t$. The goal of a bandit strategy is to maximise the sum of rewards $\sum_{t=1}^n f(\mathbf{x}_t)$ or equivalently minimise the *cumulative regret* $\sum_{t=1}^n f_\star - f(\mathbf{x}_t)$ after $n$ queries; i.e. we compete against an oracle which queries at $x_\star$ at all $t$.

Our primary distinction from the classical setting is that we have access to $M-1$ successively accurate approximations $f^{(1)}, f^{(2)}, \ldots, f^{(M-1)}$ to the payoff $f = f^{(M)}$. We refer to these approximations as fidelities. We encode the fact that fidelity $m$ approximates fidelity $M$ via the assumption, $\|f^{(M)} - f^{(m)}\|_\infty \le \zeta^{(m)}$, where $\zeta^{(1)} > \zeta^{(2)} > \cdots > \zeta^{(M)} = 0$. Each query at fidelity $m$ expends a cost $\lambda^{(m)}$ of a resource, e.g. computational effort or advertising time, where $\lambda^{(1)} < \lambda^{(2)} < \cdots < \lambda^{(M)}$. A strategy for multi-fidelity bandits is a sequence of query-fidelity pairs $\{(\mathbf{x}_t, \mathbf{m}_t)\}_{t \ge 0}$, where

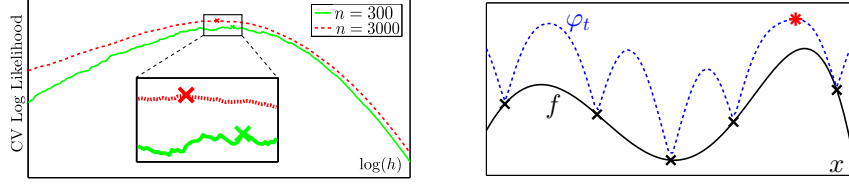

Figure 1: *Left:* Average CV log likelihood on datasets of size 300, 3000 on a synthetic KDE task. The crosses are the maxima. *Right:* Illustration of GP-UCB at time $t$. The figure shows $f(x)$ (solid black line), the UCB $\varphi_t(x)$ (dashed blue line) and queries until $t-1$ (black crosses). We query at $\mathbf{x}_t = \operatorname{argmax}_{x \in \mathcal{X}} \varphi_t(x)$ (red star).

$(\mathbf{x}_n, \mathbf{m}_n)$ could depend on the previous query-observation-fidelity tuples $\{(\mathbf{x}_t, \mathbf{y}_t, \mathbf{m}_t)\}_{t=1}^{n-1}$. Here $\mathbf{y}_t = f^{(\mathbf{m}_t)}(\mathbf{x}_t) + \epsilon$. After $n$ steps we will have queried any of the $M$ fidelities multiple times.

Some smoothness assumptions on $f^{(m)}$'s are needed to make the problem tractable. A standard in the Bayesian nonparametric literature is to use a Gaussian process (GP) prior [24] with covariance kernel $\kappa$. In this work we focus on the squared exponential (SE) $\kappa_{\sigma,h}$ and the Matérn $\kappa_{\nu,h}$ kernels as they are popularly used in practice and their theoretical properties are well studied. Writing $z = \|x - x'\|_2$, they are defined as $\kappa_{\sigma,h}(x, x') = \sigma \exp\left(-z^2/(2h^2)\right)$, $\kappa_{\nu,h}(x, x') = \frac{2^{1-\nu}}{\Gamma(\nu)} \left(\frac{\sqrt{2\nu}z}{h}\right)^\nu B_\nu\left(\frac{\sqrt{2\nu}z}{h}\right)$, where $\Gamma, B_\nu$ are the Gamma and modified Bessel functions. A convenience the GP framework offers is that posterior distributions are analytically tractable. If $f \sim \mathcal{GP}(0, \kappa)$, and we have observations $\mathcal{D}_n = \{(x_i, y_i)\}_{i=1}^n$, where $y_i = f(x_i) + \epsilon$ and $\epsilon \sim \mathcal{N}(0, \eta^2)$ is Gaussian noise, the posterior distribution for $f(x)|\mathcal{D}_n$ is also Gaussian $\mathcal{N}(\mu_n(x), \sigma_n^2(x))$ with

$$\mu_n(x) = \mathbf{k}^\top \Delta^{-1} Y, \qquad \sigma_n^2(x) = \kappa(x, x) - \mathbf{k}^\top \Delta^{-1} \mathbf{k}. \tag{1}$$

Here, $Y \in \mathbb{R}^n$ with $Y_i = y_i$, $\mathbf{k} \in \mathbb{R}^n$ with $\mathbf{k}_i = \kappa(x, x_i)$ and $\Delta = \mathbf{K} + \eta^2 I \in \mathbb{R}^{n \times n}$ where $\mathbf{K}_{i,j} = \kappa(x_i, x_j)$. In keeping with the above, we make the following assumptions on our problem.

**Assumption 1. A1**: *The functions at all fidelities are sampled from GPs,* $f^{(m)} \sim \mathcal{GP}(\mathbf{0}, \kappa)$ *for all* $m = 1, \ldots, M$. **A2**: $\|f^{(M)} - f^{(m)}\|_\infty \leq \zeta^{(m)}$ *for all* $m = 1, \ldots, M$. **A3**: $\|f^{(M)}\|_\infty \leq B$.

The purpose of **A3** is primarily to define the regret. In Remark 7, Appendix A.4 we argue that these assumptions are probabilistically valid, i.e. the latter two events occur with nontrivial probability when we sample the $f^{(m)}$'s from a GP. So a generative mechanism would keep sampling the functions and deliver them when the conditions hold true. A point $x \in \mathcal{X}$ can be queried at any of the $M$ fidelities. When we query at fidelity $m$, we observe $y = f^{(m)}(x) + \epsilon$ where $\epsilon \sim \mathcal{N}(0, \eta^2)$.

We now present our notion of cumulative regret $R(\Lambda)$ after spending capital $\Lambda$ of a resource in the multi-fidelity setting. $R(\Lambda)$ should reduce to the conventional definition of regret for any single fidelity strategy that queries only at $M^{\text{th}}$ fidelity. As only the optimum of $f = f^{(M)}$ is of interest to us, queries at fidelities less than $M$ should yield the lowest possible reward, $(-B)$ according to **A3**. Accordingly, we set the instantaneous reward $q_t$ at time to be $-B$ if $\mathbf{m}_t \neq M$ and $f^{(M)}(\mathbf{x}_t)$ if $\mathbf{m}_t = M$. If we let $r_t = f_\star - q_t$ denote the instantaneous regret, we have $r_t = f_\star + B$ if $\mathbf{m}_t \neq M$ and $f_\star - f(\mathbf{x}_t)$ if $\mathbf{m}_t = M$. $R(\Lambda)$ should also factor in the costs of the fidelity of each query. Finally, we should also receive $(-B)$ reward for any unused capital. Accordingly, we define $R(\Lambda)$ as,

$$R(\Lambda) = \Lambda f_\star - \left[\sum_{t=1}^N \lambda^{(m_t)} q_t + \left(\Lambda - \sum_{t=1}^N \lambda^{(m_t)}\right)(-B)\right] \leq 2B\Lambda_{res} + \sum_{t=1}^N \lambda^{(m_t)} r_t, \tag{2}$$

where $\Lambda_{res} = \Lambda - \sum_{t=1}^N \lambda^{(m_t)}$. Here, $N$ is the (random) number of queries at all fidelities within capital $\Lambda$, i.e. the largest $n$ such that $\sum_{t=1}^n \lambda^{(m_t)} \leq \Lambda$. According to (2) above, we wish to compete against an oracle that uses all its capital $\Lambda$ to query $x_\star$ at the $M^{\text{th}}$ fidelity. $R(\Lambda)$ is at best 0 when we follow the oracle and at most $2\Lambda B$. Our goal is a strategy that has small regret for all values of (sufficiently large) $\Lambda$, i.e. the equivalent of an anytime strategy, as opposed to a fixed time horizon strategy in the usual bandit setting. For the purpose of optimisation, we also define the *simple regret* as $S(\Lambda) = \min_t r_t = f_\star - \max_t q_t$. $S(\Lambda)$ is the difference between $f_\star$ and the best highest fidelity query (and $f_\star + B$ if we have never queried at fidelity $M$). Since $S(\Lambda) \leq \frac{1}{\Lambda} R(\Lambda)$, any strategy with asymptotic sublinear regret $\lim_{\Lambda \to \infty} \frac{1}{\Lambda} R(\Lambda) = 0$, also has vanishing simple regret.

Since, to our knowledge, this is the first attempt to formalise regret for multi-fidelity problems, the definition for $R(\Lambda)$ (2) necessitates justification. Consider a two fidelity robot gold mining problem

where the second fidelity is a real world robot trial, costing $\lambda^{(2)}$ dollars and the first fidelity is a computer simulation costing $\lambda^{(1)}$. A multi-fidelity algorithm queries the simulator to learn about the real world. But it does not collect any actual gold during a simulation; hence no reward, which according to our assumptions is $-B$. Meantime the oracle is investing this capital on the best experiment and collecting $\sim f_\star$ gold. Therefore, the regret at this time instant is $f_\star + B$. However we weight this by the cost to account for the fact that the simulation costs only $\lambda^{(1)}$. Note that lower fidelities use up capital but yield the lowest reward. The goal however, is to leverage information from these cheap queries to query prudently at the highest fidelity and obtain better regret.

That said, other multi-fidelity settings might require different definitions for $R(\Lambda)$. In online advertising, the lower fidelities (displaying ads for shorter periods) would still yield rewards. In clinical trials, the regret at the highest fidelity due to a bad treatment would be, say, a dead patient. However, a bad treatment on a simulation may not warrant large penalty. We use the definition in (2) because it is more aligned with our optimisation experiments: lower fidelities are useful to the extent that they guide search on the expensive $f^{(M)}$, but there is no reward to finding the optimum of a cheap $f^{(m)}$.

A crucial challenge for a multi-fidelity method is to not get stuck at the optimum of a lower fidelity, which is typically suboptimal for $f^{(M)}$. While exploiting information from the lower fidelities, it is also important to *explore* sufficiently at $f^{(M)}$. In our experiments we demonstrate that naive strategies which do not do so would get stuck at the optimum of a lower fidelity.

**A note on GP-UCB:** Sequential optimisation methods adopting UCB principles maintain a high probability upper bound $\varphi_t : \mathcal{X} \to \mathbb{R}$ for $f(x)$ for all $x \in \mathcal{X}$ [2]. For GP-UCB, $\varphi_t$ takes the form $\varphi_t(x) = \mu_{t-1}(x) + \beta_t^{1/2}\sigma_{t-1}(x)$ where $\mu_{t-1}, \sigma_{t-1}$ are the posterior mean and standard deviation of the GP conditioned on the previous $t-1$ queries. The key intuition is that the mean $\mu_{t-1}$ encourages an exploitative strategy – in that we want to query where we know the function is high – and the confidence band $\beta_t^{1/2}\sigma_{t-1}$ encourages an explorative strategy – in that we want to query at regions we are uncertain about $f$ lest we miss out on high valued regions. We have illustrated GP-UCB in Fig 1 and reviewed the algorithm and its theoretical properties in Appendix A.2.

## 3  MF-GP-UCB

The proposed algorithm, MF-GP-UCB, will also maintain a UCB for $f^{(M)}$ obtained via the previous queries at *all* fidelities. Denote the posterior GP mean and standard deviation of $f^{(m)}$ conditioned *only* on the previous queries at fidelity $m$ by $\mu_t^{(m)}, \sigma_t^{(m)}$ respectively (See (1)). Then define,

$$\varphi_t^{(m)}(x) = \mu_{t-1}^{(m)}(x) + \beta_t^{1/2}\sigma_{t-1}^{(m)}(x) + \zeta^{(m)}, \quad \forall m, \qquad \varphi_t(x) = \min_{m=1,\ldots,M} \varphi_t^{(m)}(x). \quad (3)$$

For appropriately chosen $\beta_t$, $\mu_{t-1}^{(m)}(x) + \beta_t^{1/2}\sigma_{t-1}^{(m)}(x)$ will upper bound $f^{(m)}(x)$ with high probability. By **A2**, $\varphi_t^{(m)}(x)$ upper bounds $f^{(M)}(x)$ for all $m$. We have $M$ such upper bounds, and their minimum $\varphi_t(x)$ gives the best bound. Our next query is at the maximiser of this UCB, $\mathbf{x}_t = \operatorname{argmax}_{x \in \mathcal{X}} \varphi_t(x)$.

Next we need to decide which fidelity to query at. Consider any $m < M$. The $\zeta^{(m)}$ conditions on $f^{(m)}$ constrain the value of $f^{(M)}$ – the confidence band $\beta_t^{1/2}\sigma_{t-1}^{(m)}$ for $f^{(m)}$ is lengthened by $\zeta^{(m)}$ to obtain confidence on $f^{(M)}$. If $\beta_t^{1/2}\sigma_{t-1}^{(m)}(\mathbf{x}_t)$ for $f^{(m)}$ is large, it means that we have not constrained $f^{(m)}$ sufficiently well at $\mathbf{x}_t$ and should query at the $m^{\text{th}}$ fidelity. On the other hand, querying indefinitely in the same region to reduce $\beta_t^{1/2}\sigma_{t-1}^{(m)}$ in that region will not help us much as the $\zeta^{(m)}$ elongation caps off how much we can learn about $f^{(M)}$ from $f^{(m)}$; i.e. even if we knew $f^{(m)}$ perfectly, we will only have constrained $f^{(M)}$ to within a $\pm\zeta^{(m)}$ band. Our algorithm captures this simple intuition. Having selected $\mathbf{x}_t$, we begin by checking at the first fidelity. If $\beta_t^{1/2}\sigma_{t-1}^{(1)}(\mathbf{x}_t)$ is smaller than a threshold $\gamma^{(1)}$, we proceed to the second fidelity. If at any stage $\beta_t^{1/2}\sigma_{t-1}^{(m)}(\mathbf{x}_t) \geq \gamma^{(m)}$ we query at fidelity $\mathbf{m}_t = m$. If we proceed all the way to fidelity $M$, we query at $\mathbf{m}_t = M$. We will discuss choices for $\gamma^{(m)}$ shortly. We summarise the resulting procedure in Algorithm 1.

Fig 2 illustrates MF-GP-UCB on a 2–fidelity problem. Initially, MF-GP-UCB is mostly exploring $\mathcal{X}$ in the first fidelity. $\beta_t^{1/2}\sigma_{t-1}^{(1)}$ is large and we are yet to constrain $f^{(1)}$ well to proceed to $f^{(2)}$. By $t = 14$, we have constrained $f^{(1)}$ around the optimum and have started querying at $f^{(2)}$ in this region.

- For $m = 1, \dots, M$:     $\mathcal{D}_0^{(m)} \leftarrow \varnothing$,    $(\mu_0^{(m)}, \sigma_0^{(m)}) \leftarrow (\mathbf{0}, \kappa^{1/2})$.
- for $t = 1, 2, \dots$
  1. $\mathbf{x}_t \leftarrow \mathrm{argmax}_{x \in \mathcal{X}} \; \varphi_t(x)$.      (See Equation (3))
  2. $\mathbf{m}_t = \min_m \{ m \, | \beta_t^{1/2} \sigma_{t-1}^{(m)}(\mathbf{x}_t) \geq \gamma^{(m)} \;$ or $\; m = M \}$.      (See Appendix B, C for $\beta_t$)
  3. $\mathbf{y}_t \leftarrow$ Query $f^{(\mathbf{m}_t)}$ at $\mathbf{x}_t$.
  4. Update $\mathcal{D}_t^{(\mathbf{m}_t)} \leftarrow \mathcal{D}_{t-1}^{(\mathbf{m}_t)} \cup \{(\mathbf{x}_t, \mathbf{y}_t)\}$. Obtain $\mu_t^{(\mathbf{m}_t)}, \sigma_t^{(\mathbf{m}_t)}$ conditioned on $\mathcal{D}_t^{(\mathbf{m}_t)}$ (See (1)).

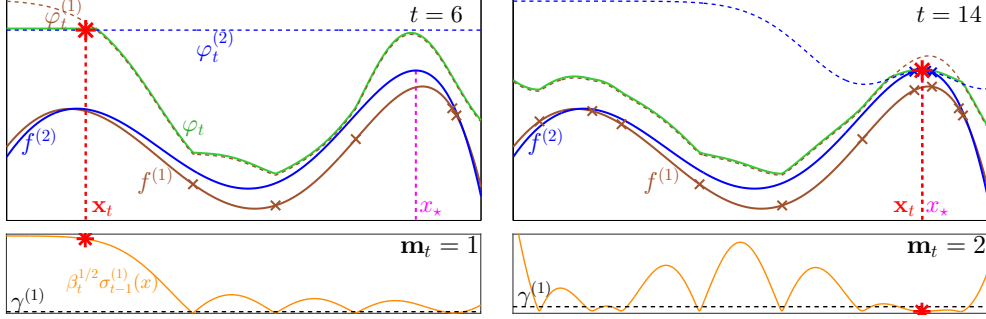

Figure 2:   Illustration of MF-GP-UCB for a 2-fidelity problem initialised with 5 random points at the first fidelity. In the top figures, the solid lines in brown and blue are $f^{(1)}, f^{(2)}$ respectively, and the dashed lines are $\varphi_t^{(1)}, \varphi_t^{(2)}$. The solid green line is $\varphi_t = \min(\varphi_t^{(1)}, \varphi_t^{(2)})$. The small crosses are queries from 1 to $t-1$ and the red star is the maximiser of $\varphi_t$, i.e. the next query $\mathbf{x}_t$. $x_\star$, the optimum of $f^{(2)}$ is shown in magenta. In the bottom figures, the solid orange line is $\beta_t^{1/2} \sigma_{t-1}^{(1)}$ and the dashed black line is $\gamma^{(1)}$. When $\beta_t^{1/2} \sigma_{t-1}^{(1)}(\mathbf{x}_t) \leq \gamma^{(1)}$ we play at fidelity $\mathbf{m}_t = 2$ and otherwise at $\mathbf{m}_t = 1$. See Fig. 6 in Appendix B for an extended simulation.

Notice how $\varphi_t^{(2)}$ dips to change $\varphi_t$ in this region. MF-GP-UCB has identified the maximum with just 3 queries to $f^{(2)}$. In Appendix B we provide an extended simulation and discuss further insights.

Finally, we make an essential observation. The posterior for any $f^{(m)}(x)$ conditioned on previous queries at *all* fidelities is not Gaussian due to the $\zeta^{(m)}$ constraints (**A2**). However, $|f^{(m)}(x) - \mu_{t-1}^{(m)}(x)| < \beta_t^{1/2} \sigma_{t-1}^{(m)}(x)$ holds with high probability, since, by conditioning only on queries at the $m^{\text{th}}$ fidelity we have Gaussianity for $f^{(m)}(x)$. Next we summarise our main theoretical contributions.

## 4   Summary of Theoretical Results

For pedagogical reasons we present our results for the $M = 2$ case. Appendix C contains statements and proofs for general $M$. We also ignore constants and polylog terms when they are dominated by other terms. $\lesssim, \asymp$ denote inequality and equality ignoring constants. We begin by defining the *Maximum Information Gain* (MIG) which characterises the statistical difficulty of GP bandits.

**Definition 2.** *(Maximum Information Gain) Let $f \sim \mathcal{GP}(\mathbf{0}, \kappa)$. Consider any $A \subset \mathbb{R}^d$ and let $\widetilde{A} = \{x_1, \dots, x_n\} \subset A$ be a finite subset. Let $f_{\widetilde{A}}, \epsilon_{\widetilde{A}} \in \mathbb{R}^n$ be such that $(f_{\widetilde{A}})_i = f(x_i)$, $(\epsilon_{\widetilde{A}})_i \sim \mathcal{N}(0, \eta^2)$, and $y_{\widetilde{A}} = f_{\widetilde{A}} + \epsilon_{\widetilde{A}}$. Let $I$ denote the Shannon Mutual Information. The Maximum Information Gain of $A$ is $\Psi_n(A) = \max_{\widetilde{A} \subset A, |\widetilde{A}| = n} I(y_{\widetilde{A}}; f_{\widetilde{A}})$.*

The MIG, which depends on the kernel $\kappa$ and the set $A$, is an important quantity in our analysis. For a given $\kappa$, it typically scales with the volume of $A$; i.e. if $A = [0, r]^d$ then $\Psi_n(A) \in \mathcal{O}(r^d \Psi_n([0,1]^d))$. For the SE kernel, $\Psi_n([0,1]^d) \in \mathcal{O}((\log(n))^{d+1})$ and for Matérn, $\Psi_n([0,1]^d) \in \mathcal{O}(n^{\frac{d(d+1)}{2\nu + d(d+1)}})$ [28].

Recall, $N$ is the (random) number of queries by a multi-fidelity strategy within capital $\Lambda$ at either fidelity. Let $n_\Lambda = \lfloor \Lambda / \lambda^{(2)} \rfloor$ be the (non-random) number of queries by a single fidelity method operating only at the second fidelity. As $\lambda^{(1)} < \lambda^{(2)}$, $N$ could be large for an arbitrary multi-fidelity method. However, our analysis reveals that for MF-GP-UCB, $N$ is on the order of $n_\Lambda$.

Fundamental to the 2-fidelity problem is the set $\mathcal{X}_g = \{x \in \mathcal{X}; f_\star - f^{(1)}(x) \leq \zeta^{(1)}\}$. $\mathcal{X}_g$ is a high valued region for $f^{(2)}(x)$: for all $x \in \mathcal{X}_g$, $f^{(2)}(x)$ is at most $2\zeta^{(1)}$ away from the optimum. More interestingly, when $\zeta^{(1)}$ is small, i.e. when $f^{(1)}$ is a good approximation to $f^{(2)}$, $\mathcal{X}_g$ will be much smaller than $\mathcal{X}$. This is precisely the target domain for this research. For instance, in the robot gold mining example, a cheap computer simulator can be used to eliminate several bad policies and we could reserve the real world trials for the promising candidates. If a multi-fidelity strategy were to use the second fidelity queries only in $\mathcal{X}_g$, then the regret will only have $\Psi_n(\mathcal{X}_g)$ dependence after $n$ high fidelity queries. In contrast, a strategy that only operates at the highest fidelity (e.g. GP-UCB) will have $\Psi_n(\mathcal{X})$ dependence. In the scenario described above $\Psi_n(\mathcal{X}_g) \ll \Psi_n(\mathcal{X})$, and the multi-fidelity strategy will have significantly better regret than a single fidelity strategy. MF-GP-UCB roughly achieves this goal. In particular, we consider a slightly inflated set $\widetilde{\mathcal{X}}_{g,\rho} = \{x \in \mathcal{X}; f_\star - f^{(1)}(x) \leq \zeta^{(1)} + \rho\gamma^{(1)}\}$, of $\mathcal{X}_g$ where $\rho > 0$. The following result which characterises the regret of MF-GP-UCB in terms of $\widetilde{\mathcal{X}}_{g,\rho}$ is the main theorem of this paper.

**Theorem 3** (Regret of MF-GP-UCB – Informal). *Let $\mathcal{X} = [0, r]^d$ and $f^{(1)}, f^{(2)} \sim \mathcal{GP}(\mathbf{0}, \kappa)$ satisfy Assumption 1. Pick $\delta \in (0, 1)$ and run MF-GP-UCB with $\beta_t \asymp d\log(t/\delta)$. Then, with probability $> 1 - \delta$, for sufficiently large $\Lambda$ and for all $\alpha \in (0, 1)$, there exists $\rho$ depending on $\alpha$ such that,*

$$R(\Lambda) \lesssim \lambda^{(2)}\sqrt{n_\Lambda \beta_{n_\Lambda} \Psi_{n_\Lambda}(\widetilde{\mathcal{X}}_{g,\rho})} + \lambda^{(1)}\sqrt{n_\Lambda \beta_{n_\Lambda} \Psi_{n_\Lambda}(\mathcal{X})} + \lambda^{(2)}\sqrt{n_\Lambda^\alpha \beta_{n_\Lambda} \Psi_{n_\Lambda^\alpha}(\mathcal{X})} + \lambda^{(1)}\xi_{n, \widetilde{\mathcal{X}}_{g,\rho}, \gamma^{(1)}}$$

As we will explain shortly, the latter two terms are of lower order. It is instructive to compare the above rates against that for GP-UCB (see Theorem 4, Appendix A.2). By dropping the common and subdominant terms, the rate for MF-GP-UCB is $\lambda^{(2)}\Psi_{n_\Lambda}^{1/2}(\widetilde{\mathcal{X}}_{g,\rho}) + \lambda^{(1)}\Psi_{n_\Lambda}^{1/2}(\mathcal{X})$ whereas for GP-UCB it is $\lambda^{(2)}\Psi_{n_\Lambda}^{1/2}(\mathcal{X})$. When $\lambda^{(1)} \ll \lambda^{(2)}$ and $\text{vol}(\widetilde{\mathcal{X}}_{g,\rho}) \ll \text{vol}(\mathcal{X})$ the rates for MF-GP-UCB are very appealing. When the approximation worsens ($\mathcal{X}_g, \widetilde{\mathcal{X}}_{g,\rho}$ become larger) and the costs $\lambda^{(1)}, \lambda^{(2)}$ become comparable, the bound for MF-GP-UCB decays gracefully. In the worst case, MF-GP-UCB is never worse than GP-UCB up to constant terms. Intuitively, the above result states that MF-GP-UCB explores the entire $\mathcal{X}$ using $f^{(1)}$ but uses "most" of its queries to $f^{(2)}$ inside $\widetilde{\mathcal{X}}_{g,\rho}$.

Now let us turn to the latter two terms in the bound. The third term is the regret due to the second fidelity queries outside $\widetilde{\mathcal{X}}_{g,\rho}$. We are able to show that the number of such queries is $\mathcal{O}(n_\Lambda^\alpha)$ for all $\alpha > 0$ for an appropriate $\rho$. This *strong* result is only possible in the multi-fidelity setting. For example, in GP-UCB the best bound you can achieve on the number of plays on a suboptimal set is $\mathcal{O}(n_\Lambda^{1/2})$ for the SE kernel and worse for the Matérn kernel. The last term is due to the first fidelity plays inside $\widetilde{\mathcal{X}}_{g,\rho}$ and it scales with $\text{vol}(\widetilde{\mathcal{X}}_{g,\rho})$ and polylogarithmically with $n$, both of which are small. However, it has a $1/\text{poly}(\gamma^{(1)})$ dependence which could be bad if $\gamma^{(1)}$ is too small: intuitively, if $\gamma^{(1)}$ is too small then you will wait for a long time in step 2 of Algorithm 1 for $\beta_t^{1/2}\sigma_{t-1}^{(1)}$ to decrease without proceeding to $f^{(2)}$, incurring large regret ($f_\star + B$) in the process. Our analysis reveals that an optimal choice for the SE kernel scales $\gamma^{(1)} \asymp (\lambda^{(1)}\zeta^{(1)}/(t\lambda^{(2)}))^{1/(d+2)}$ at time $t$. However this is of little practical use as the leading constant depends on several problem dependent quantities such as $\Psi_n(\mathcal{X}_g)$. In Section 5 we describe a heuristic to set $\gamma^{(m)}$ which worked well in our experiments.

Theorem 3 can be generalised to cases where the kernels $\kappa^{(m)}$ and observation noises $\eta^{(m)}$ are different at each fidelity. The changes to the proofs are minimal. In fact, our practical implementation uses different kernels. As with any nonparametric method, our algorithm has exponential dependence on dimension. This can be alleviated by assuming additional structure in the problem [8, 15]. Finally, we note that the above rates translate to bounds on the simple regret $S(\Lambda)$ for optimisation.

## 5 Implementation Details

Our implementation uses some standard techniques in Bayesian optimisation to learn the kernel such as initialisation with random queries and periodic marginal likelihood maximisation. The above techniques might be already known to a reader familiar with the BO literature. We have elaborated these in Appendix B but now focus on the $\gamma^{(m)}, \zeta^{(m)}$ parameters of our method.

Algorithm 1 assumes that the $\zeta^{(m)}$'s are given with the problem description, which is hardly the case in practice. In our implementation, instead of having to deal with $M - 1$, $\zeta^{(m)}$ values we set $(\zeta^{(1)}, \zeta^{(2)}, \ldots, \zeta^{(M-1)}) = ((M-1)\zeta, (M-2)\zeta, \ldots, \zeta)$ so we only have one value $\zeta$. This for

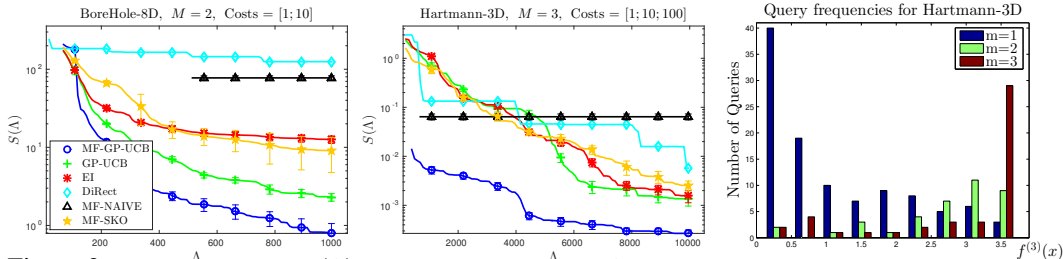

Figure 3: The simple regret $S(\Lambda)$ against the spent capital $\Lambda$ on synthetic functions. The title states the function, its dimensionality, the number of fidelities and the costs we used for each fidelity in the experiment. All curves barring DiRect (which is a deterministic), were produced by averaging over 20 experiments. The error bars indicate one standard error. See Figures 8, 9 10 in Appendix D for more synthetic results. The last panel shows the number of queries at different function values at each fidelity for the Hartmann-3D example.

instance, is satisfied if $\|f^{(m)} - f^{(m-1)}\|_\infty \leq \zeta$ which is stronger than Assumption **A2**. Initially, we start with small $\zeta$. Whenever we query at any fidelity $m > 1$ we also check the posterior mean of the $(m-1)^{\text{th}}$ fidelity. If $|f^{(m)}(\mathbf{x}_t) - \mu_{t-1}^{(m-1)}(\mathbf{x}_t)| > \zeta$, we query again at $\mathbf{x}_t$, but at the $(m-1)^{\text{th}}$ fidelity. If $|f^{(m)}(\mathbf{x}_t) - f^{(m-1)}(\mathbf{x}_t)| > \zeta$, we update $\zeta$ to twice the violation. To set $\gamma^{(m)}$'s we use the following intuition: if the algorithm, is stuck at fidelity $m$ for too long then $\gamma^{(m)}$ is probably too small. We start with small values for $\gamma^{(m)}$. If the algorithm does not query above the $m^{\text{th}}$ fidelity for more than $\lambda^{(m+1)}/\lambda^{(m)}$ iterations, we double $\gamma^{(m)}$. We found our implementation to be fairly robust even recovering from fairly bad approximations at the lower fidelities (see Appendix D.3).

## 6 Experiments

We compare MF-GP-UCB to the following methods. **Single fidelity methods:** GP-UCB; EI: the expected improvement criterion for BO [13]; DiRect: the dividing rectangles method [12]. **Multi-fidelity methods:** MF-NAIVE: a naive baseline where we use GP-UCB to query at the *first* fidelity a large number of times and then query at the last fidelity at the points queried at $f^{(1)}$ in decreasing order of $f^{(1)}$-value; MF-SKO: the multi-fidelity sequential kriging method from [11]. Previous works on multi-fidelity methods (including MF-SKO) had not made their code available and were not straightforward to implement. Hence, we could not compare to all of them. We discuss this more in Appendix D along with some other single and multi-fidelity baselines we tried but excluded in the comparison to avoid clutter in the figures. In addition, we also detail the design choices and hyper-parameters for all methods in Appendix D.

**Synthetic Examples:** We use the Currin exponential ($d = 2$), Park ($d = 4$) and Borehole ($d = 8$) functions in $M = 2$ fidelity experiments and the Hartmann functions in $d = 3$ and 6 with $M = 3$ and 4 fidelities respectively. The first three are taken from previous multi-fidelity literature [32] while we tweaked the Hartmann functions to obtain the lower fidelities for the latter two cases. We show the simple regret $S(\Lambda)$ against capital $\Lambda$ for the Borehole and Hartmann-3D functions in Fig. 3 with the rest deferred to Appendix D due to space constraints. MF-GP-UCB outperforms other methods. Appendix D also contains results for the cumulative regret $R(\Lambda)$ and the formulae for these functions.

A common occurrence with MF-NAIVE was that once we started querying at fidelity $M$, the regret barely decreased. The diagnosis in all cases was the same: it was stuck around the maximum of $f^{(1)}$ which is suboptimal for $f^{(M)}$. This suggests that while we have cheap approximations, the problem is by no means trivial. As explained previously, it is also important to "explore" at the higher fidelities to achieve good regret. The efficacy of MF-GP-UCB when compared to single fidelity methods is that it confines this exploration to a small set containing the optimum. In our experiments we found that MF-SKO did not consistently beat other single fidelity methods. Despite our best efforts to reproduce this (and another) multi-fidelity method, we found them to be quite brittle (Appendix D.1).

The third panel of Fig. 3 shows a histogram of the number of queries at each fidelity after 184 queries of MF-GP-UCB, for different ranges of $f^{(3)}(x)$ for the Hartmann-3D function. Many of the queries at the low $f^{(3)}$ values are at fidelity 1, but as we progress they decrease and the second fidelity queries increase. The third fidelity dominates very close to the optimum but is used sparingly elsewhere. This corroborates the prediction in our analysis that MF-GP-UCB uses low fidelities to explore and successively higher fidelities at promising regions to zero in on $x_\star$. (Also see Fig. 6, Appendix B.)

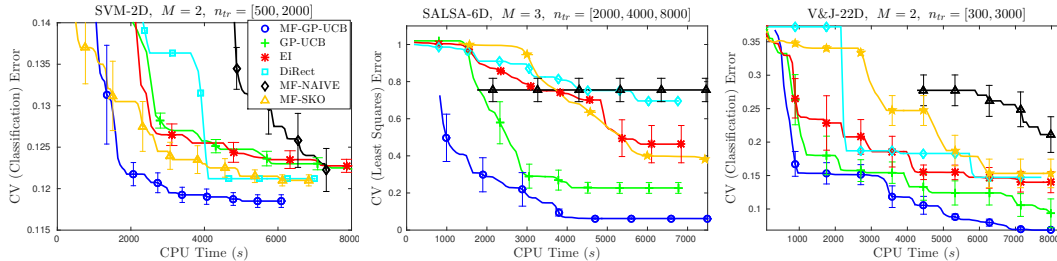

Figure 4: Results on the hyper-parameter tuning experiments. The title states the experiment, dimensionality (number of hyperparameters) and training set size at each fidelity. All curves were produced by averaging over 10 experiments. The error bars indicate one standard error. The lengths of the curves are different in time as we ran each method for a pre-specified number of iterations and they concluded at different times.

**Real Experiments:** We present results on three hyper-parameter tuning tasks (results in Fig. 4), and a maximum likelihood inference task in Astrophysics (Fig. 5). We compare methods on computation time since that is the "cost" in all experiments. We include the processing time for each method in the comparison (i.e. the cost of determining the next query).

**Classification using SVMs (SVM):** We trained an SVM on the magic gamma dataset using the SMO algorithm to an accuracy of $10^{-12}$. The goal is to tune the kernel bandwidth and the soft margin coefficient in the ranges $(10^{-3}, 10^1)$ and $(10^{-1}, 10^5)$ respectively on a dataset of size 2000. We set this up as a $M = 2$ fidelity experiment with the entire training set at the second fidelity and 500 points at the first. Each query was 5-fold cross validation on these training sets.

**Regression using Additive Kernels (SALSA):** We used the regression method from [14] on the 4-dimensional coal power plant dataset. We tuned the 6 hyper-parameters –the regularisation penalty, the kernel scale and the kernel bandwidth for each dimension– each in the range $(10^{-3}, 10^4)$ using 5-fold cross validation. This experiment used $M = 3$ and $2000, 4000, 8000$ points at each fidelity.

**Viola & Jones face detection (V&J):** The V&J classifier [31], which uses a cascade of weak classifiers, is a popular method for face detection. To classify an image, we pass it through each classifier. If at any point the classifier score falls below a threshold, the image is classified negative. If it passes through the cascade, then it is classified positive. One of the more popular implementations comes with OpenCV and uses a cascade of 22 weak classifiers. The threshold values in OpenCV are pre-set based on some heuristics and there is no reason to think they are optimal for a given face detection task. The goal is to tune these 22 thresholds by optimising for them over a training set. We modified the OpenCV implementation to take in the thresholds as parameters. As our domain $\mathcal{X}$ we chose a neighbourhood around the configuration used in OpenCV. We set this up as a $M = 2$ fidelity experiment where the second fidelity used 3000 images from the V&J face database and the first used 300. Interestingly, on an independent test set, the configurations found by MF-GP-UCB consistently achieved over $90\%$ accuracy while the OpenCV configuration achieved only $87.4\%$ accuracy.

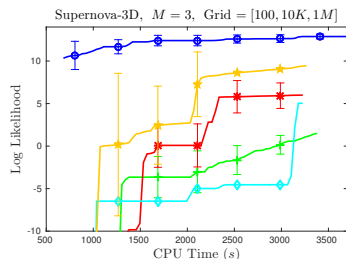

Figure 5: Results on the supernova inference problem. The $y$-axis is the log likelihood so higher is better. MF-NAIVE is not visible as it performed very poorly.

**Type Ia Supernovae:** We use Type Ia supernovae data [7] for maximum likelihood inference on 3 cosmological parameters, the Hubble constant $H_0 \in (60, 80)$, the dark matter and dark energy fractions $\Omega_M, \Omega_\Lambda \in (0, 1)$. Unlike typical parametric maximum likelihood problems, the likelihood is only available as a black-box. It is computed using the Robertson–Walker metric which requires a one dimensional numerical integration for each sample in the dataset. We set this up as a $M = 3$ fidelity task. The goal is to maximise the likelihood at the third fidelity where the integration was performed using the trapezoidal rule on a grid of size $10^6$. For the first and second fidelities, we used grids of size $10^2, 10^4$ respectively. The results are given in Fig. 5.

**Conclusion:** We introduced and studied the multi-fidelity bandit under Gaussian Process assumptions. We present, to our knowledge, the first formalism of regret and the first theoretical results in this setting. They demonstrate that MF-GP-UCB explores the space via cheap lower fidelities, and leverages the higher fidelities on successively smaller regions hence achieving better regret than single fidelity strategies. Experimental results demonstrate the efficacy of our method.

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
