[Supplementary Material]

# Appendix

## A  Some Ancillary Material

### A.1  Table of Notations

| | |
|---|---|
| $M$ | The number of fidelities. |
| $f, f^{(m)}$ | The payoff function and its $m^{\text{th}}$ fidelity approximation. $f^{(M)} = f$. |
| $\lambda^{(m)}$ | The cost for querying at fidelity $m$. |
| $\mathcal{X}$ | The domain over which we are optimising $f$. |
| $x_\star, f_\star$ | The optimum point and value of the $M^{\text{th}}$ fidelity function. |
| $\overline{A}$ | The complement of a set $A \subset \mathcal{X}$. $\overline{A} = \mathcal{X} \backslash A$. |
| $|A|$ | The cardinality of a set $A \subset \mathcal{X}$ if it is countable. |
| $\vee, \wedge$ | Logical *Or* and *And* respectively. |
| $\lesssim, \gtrsim, \asymp$ | Inequalities and equality ignoring constant terms. |
| $q_t, r_t$ | The instantaneous reward and regret respectively. $q_t = f^{(M)}(\mathbf{x}_t)$ if $\mathbf{m}_t = M$ and $-B$ if $\mathbf{m}_t \neq M$. $r_t = f_\star - q_t$. |
| $R(\Lambda)$ | The cumulative regret after spending capital $\Lambda$. See equation (2). |
| $S(\Lambda)$ | The simple regret after spending capital $\Lambda$. See second paragraph under equation (2). |
| $\zeta^{(m)}$ | A bound on the maximum difference between $f^{(m)}$ and $f^{(M)}$, $\|f^{(M)} - f^{(m)}\|_\infty \leq \zeta^{(m)}$. |
| $\mu_t^{(m)}$ | The mean of the $m^{\text{th}}$ fidelity GP $f^{(m)}$ conditioned on $\mathcal{D}_t^{(m)}$ at time $t$. |
| $\kappa_t^{(m)}$ | The covariance of the $m^{\text{th}}$ fidelity GP $f^{(m)}$ conditioned on $\mathcal{D}_t^{(m)}$ at time $t$. |
| $\sigma_t^{(m)}$ | The standard deviatiation of the $m^{\text{th}}$ fidelity GP $f^{(m)}$ conditioned on $\mathcal{D}_t^{(m)}$ at time $t$. |
| $\mathbf{x}_t, \mathbf{y}_t$ | The queried point and observation at time $t$. |
| $\mathbf{m}_t$ | The queried fidelity at time $t$. |
| $\mathcal{D}_n^{(m)}$ | The set of queries at the $m^{\text{th}}$ fidelity until time $n$ $\{(\mathbf{x}_t, \mathbf{y}_t)\}_{t:\mathbf{m}_t=m}$. |
| $\beta_t$ | The coefficient trading off exploration and exploitation in the UCB. See Theorem 10. |
| $\varphi_t^{(m)}(x)$ | The upper confidence bound (UCB) provided by the $m^{\text{th}}$ fidelity on $f^{(M)}(x)$. $\varphi_t^{(m)}(x) = \mu_{t-1}^{(m)}(x) + \beta_t^{1/2}\sigma_{t-1}^{(m)}(x) + \zeta^{(m)}$. |
| $\varphi_t(x)$ | The combined UCB provided by all fidelities on $f^{(M)}(x)$. $\varphi_t(x) = \min_m \varphi_t^{(m)}(x)$. |
| $\gamma^{(m)}$ | The parameter in MF-GP-UCB for switching from the $m^{\text{th}}$ fidelity to the $(m+1)^{\text{th}}$. |
| $\tilde{R}_n$ | The cumulative regret for the queries after $n$ rounds, $\tilde{R}_n = \sum_{t=1}^{n} \lambda^{(m_t)} r_t$. |
| $T_n^{(m)}(A)$ | The number of queries at fidelity $m$ in subset $A \subset \mathcal{X}$ until time $n$. |
| $T_n^{(>m)}(A)$ | The number of queries at fidelities greater than $m$ in any subset $A \subset \mathcal{X}$ until time $n$. |
| $n_\Lambda$ | Number of plays by a strategy querying only at fidelity $M$ within capital $\Lambda$. $n_\Lambda = \lfloor \Lambda/\lambda^{(M)} \rfloor$. |
| $\Psi_n(A)$ | The maximum information gain of a set $A \subset \mathcal{X}$ after $n$ queries in $A$. See Definition 2. |
| $\mathcal{X}^{(m)}$ | $(\mathcal{X}^{(m)})_{m=1}^{M}$ is an entirely problem dependent partitioning of $\mathcal{X}$. See Equation (5). |
| $\mathcal{H}_\tau^{(m)}$ | $(\mathcal{H}_\tau^{(m)})_{m=1}^{M}$ are partitionings of $\mathcal{X}$. See Equation (5). The analysis of MF-GP-UCB hinges on these partitionings. |
| $\mathcal{H}_{\tau,n}^{(m)}$ | An additional $n$-dependent inflation of $\mathcal{H}_\tau^{(m)}$. See paragraph under equation (5). |
| $\widehat{\mathcal{H}}_\tau^{(m)}, \widecheck{\mathcal{H}}_\tau^{(m)}$ | The arms "above"/"below" $\mathcal{H}_\tau^{(m)}$. $\widehat{\mathcal{H}}_\tau^{(m)} = \bigcup_{\ell=m+1}^{M} \mathcal{H}_\tau^{(\ell)}$, $\widecheck{\mathcal{H}}_\tau^{(m)} = \bigcup_{\ell=1}^{m-1} \mathcal{H}_\tau^{(\ell)}$. |
| $\mathcal{X}_g, \mathcal{X}_b$ | The good set and bad sets for $M = 2$ fidelity problems. $\mathcal{X}_g = \mathcal{X}^{(2)}$ and $\mathcal{X}_b = \mathcal{X}^{(1)}$. |
| $\widetilde{\mathcal{X}}_{g,\rho}, \widetilde{\mathcal{X}}_{b,\rho}$ | The inflations of $\mathcal{X}_g, \mathcal{X}_b$ for MF-GP-UCB. $\widetilde{\mathcal{X}}_{g,\rho} = \{x; f_\star - f^{(1)}(x) \leq \zeta^{(1)} + \rho\gamma\}$, and $\ddot{\mathcal{X}}_{b,\tau} = \mathcal{X} \backslash \ddot{\mathcal{X}}_{g,\tau}$. |
| $\Omega_\varepsilon(A)$ | The $\varepsilon$–covering number of a subset $A \subset \mathcal{X}$ in the $\|\cdot\|_2$ metric. |

### A.2  Review of GP-UCB

The following bounds the regret $R_n$ for the GP-UCB algorithm of Srinivas et al. [28] after $n$ time steps. The algorithm is given in Algorithm 2.

**Theorem 4.** *(Theorems 2 in [28]) Let $f \sim \mathcal{GP}(\mathbf{0}, \kappa)$, $f : \mathcal{X} \to \mathbb{R}$ and $\kappa$ satisfy Assumption 8. At each query, we have noisy observations $y = f(x) + \epsilon$ where $\epsilon \sim \mathcal{N}(0, \eta^2)$. Denote $C_1 = 8/\log(1 + \eta^{-2})$. Pick $\delta \in (0, 1)$. If $\mathcal{X} = [0, r]^d$, run GP-UCB with $\beta_t = 2\log\left(\frac{2\pi^2 t^2}{3\delta}\right) + 2d\log\left(t^2 bdr\sqrt{\frac{4ad}{\delta}}\right)$. Then,*

$$\mathbb{P}\left(\forall n \geq 1, \ R_n \leq \sqrt{C_1 n \beta_n \Psi_n(\mathcal{X})} + 2\right) \geq 1 - \delta$$

*Here $\Psi_n(\mathcal{X})$ is the Maximum Information Gain of $\mathcal{X}$ after $n$ queries (see Definition 2).*

---

**Algorithm 2** GP-UCB

**Input:** kernel $\kappa$.
For $t = 1, 2 \ldots$
- $\mathcal{D}_0 \leftarrow \varnothing$, $(\mu_0, \sigma_0^2) \leftarrow (\mathbf{0}, \kappa)$.
- $(\mu_0, \kappa_0) \leftarrow (\mathbf{0}, \kappa)$
- **for** $t = 1, 2, \ldots$
    1. $\mathbf{x}_t \leftarrow \operatorname{argmax}_{x \in \mathcal{X}} \mu_{t-1}(x) + \beta_t^{1/2} \sigma_{t-1}(x)$
    2. $\mathbf{y}_t \leftarrow$ Query $f$ at $\mathbf{x}_t$.
    3. $\mathcal{D}_t = \mathcal{D}_{t-1} \cup \{(\mathbf{x}_t, \mathbf{y}_t)\}$.
    4. Perform Bayesian posterior updates to obtain $\mu_t, \sigma_t$ (See Equation (1)).

---

### A.3 More Related Work

Agarwal et al. [1] derive oracle inequalities for hyper-parameter tuning with ERM under computational budgets. Our setting is more general as it applies to any bandit optimisation task. Sabharwal et al. [26] present a UCB based idea for tuning hyper-parameters with incremental data allocation. However, their theoretical results are for an idealised non-realisable algorithm. Cutler et al. [5] study reinforcement learning with multi-fidelity simulators by treating each fidelity as a Markov Decision Process. Finally, Zhang and Chaudhuri [33] study active learning when there is access to a cheap weak labeler and an expensive strong labeler. All the work above study problems different to optimisation. Further, none of them are in the bandit setting where there is a price for exploration.

### A.4 Some Ancillary Results

We will use the following results in our analysis. The first is a standard Gaussian concentration result and the second is an expression for the Information Gain in a GP from Srinivas et al. [28].

**Lemma 5** (Gaussian Concentration). *Let $Z \sim \mathcal{N}(0, 1)$. Then $\mathbb{P}(Z > \epsilon) \leq \frac{1}{2}\exp(-\epsilon^2/2)$.*

**Lemma 6** (Mutual Information in GP, [28] Lemma 5.3). *Let $f \sim \mathcal{GP}(\mathbf{0}, \kappa)$, $f : \mathcal{X} \to \mathbb{R}$ and we observe $y = f(x) + \epsilon$ where $\epsilon \sim \mathcal{N}(0, \eta^2)$. Let $A$ be a finite subset of $\mathcal{X}$ and $f_A, y_A$ be the function values and observations on this set respectively. Using the basic Gaussian properties they show that the mutual information $I(y_A; f_A)$ is,*

$$I(y_A; f_A) = \frac{1}{2}\sum_{t=1}^{n} \log(1 + \eta^{-2}\sigma_{t-1}^2(x_t)).$$

*where $\sigma_{t-1}^2$ is the posterior variance after observing the first $t - 1$ points.*

We conclude this section with the following comment on our assumptions in Section 2.

**Remark 7** (**Validity of the Assumptions A1, A2, A3**). It is sufficient to show that when the functions $f^{(m)}$ are sampled from $\mathcal{GP}(\mathbf{0}, \kappa)$, the latter constraints, i.e. $\|f^{(M)}\|_\infty \leq B$ and $\|f^{(M)} - f^{(m)}\|_\infty \leq \zeta^{(m)} \ \forall m$, occur with positive probability. Then, a generative mechanism would repeatedly sample the $f^{(m)}$'s from the GP and output them when the constraints are satisfied. The claim is true for well behaved kernels. For instance, using Assumption 8 (Appendix C) we can establish a high probability bound on the Lipschitz constant of the GP sample $f^{(M)}$. Since for

a given $x \in \mathcal{X}$, $\mathbb{P}(-B < f^{(M)}(x) < 0)$ is positive we just need to make sure that the Lipschitz constant is not larger than $B/\mathrm{diam}(\mathcal{X})$. This bounds $\|f^{(M)}\|_{\infty} < B$. For the latter constraint, since $f^{(M)} - f^{(m)} \sim \mathcal{GP}(\mathbf{0}, 2\kappa)$ is also a GP, the argument follows in an essentially similar fashion.

## B  Some Details on MF-GP-UCB

### An Extended Simulation

In Figure 6 we provide an extended version of the simulation of Fig. 2 for a 2 fidelity example. Read the caption under the simulation for more details.

### More Implementation Details

**Data dependent prior:** In our experiments, following recommendations in Brochu et al. [3] all GP methods were initialised with uniform random queries using an initialisation capital $\Lambda_0$. For single fidelity methods, we used it at the $M^{\text{th}}$ fidelity, whereas for MF-GP-UCB we used $\Lambda_0/2$ at fidelity 1 and $\Lambda_0/2$ at fidelity 2. After initialising the kernel in this manner, we update the kernel every 25 iterations of the method by maximising the GP marginal likelihood.

**Choice of $\beta_t$:** $\beta_t$, as specified in Theorems 4, 10 has unknown constants and tends to be too conservative in practice. Following Kandasamy et al. [15] we use $\beta_t = 0.2d\log(2t)$ which captures the dominant dependencies on $d$ and $t$.

**Initial $\zeta$, $\gamma$:** We set both $\zeta, \gamma$ to $1\%$ of the range of initial queries and update them as explained in the main text.

**Maximising $\varphi_t$:** To determine $\mathbf{x}_t$ we maximised $\varphi_t$ using DiRect [12]. For other GP methods, the EI, PI, GP-UCB acquisition functions were also maximised using DiRect.

MF-GP-UCB was fairly robust to the above choices except when $\Lambda_0$ was set too low in which case, all GP methods performed poorly on some experiments.

## C  Theoretical Analysis

In this section we present our main theoretical results. While it is self contained, the reader will benefit from first reading the more intuitive discussion in Section 4. The goal in this section is to bound $R(\Lambda)$ for MF-GP-UCB . Recall,

$$
\begin{aligned}
R(\Lambda) =\ & \Lambda f_\star - \sum_{t=1}^{N} \lambda^{(m_t)} q_t - \left(\Lambda - \sum_{t=1}^{N} \lambda^{(m_t)}\right)(-B) \\
=\ & \underbrace{\left(\Lambda - \sum_{t=1}^{N} \lambda^{(m_t)}\right)(f_\star + B)}_{\tilde{r}(\Lambda)} + \underbrace{\sum_{t=1}^{N} \lambda^{(m_t)} r_t}_{\tilde{R}(\Lambda)},
\end{aligned}
$$

where $N$ is the random number of plays within capital $\Lambda$ and $q_t, r_t$ are the instantaneous reward and regret as defined in Section 2. The first term $\tilde{r}(\Lambda)$ is the residual quantity. It is an artefact of the fact that after the $(N+1)^{\text{th}}$ query, the spent capital would have exceeded $\Lambda$. It can be bounded by $\tilde{r}(\Lambda) \leq 2B\lambda^{(M)}$ which is typically small. Our analysis will mostly be dealing with the latter term $\tilde{R}(\Lambda)$ for which we will first bound the quantity $\tilde{R}_n = \sum_{t=1}^{n} \lambda^{(m_t)} r_t$ after $n$ time steps in terms of $n$. Then, we will bound the random number of plays $N$ within principal $\Lambda$. While $N \leq \lfloor \Lambda/\lambda^{(1)} \rfloor$ is a trivial bound, this will be too loose for our purpose. In fact, we will show that after a sufficiently large number of time steps $n$, with high probability the number of plays at fidelities lower than $M$ will be sub-linear in $n$. Hence $N \in \mathcal{O}(n_\Lambda)$ where $n_\Lambda = \lfloor \Lambda/\lambda^{(M)} \rfloor$ is the number of plays by any algorithm that operates only at the highest fidelity.

Our strategy to bound $\tilde{R}_n$ will be to identify a (possibly disconnected) measurable region of the space $\mathcal{Z}$ which contains $x_\star$ and has high value for the payoff function $f^{(M)}(x)$. $\mathcal{Z}$ will be determined by

Figure 6: Illustration of MF-GP-UCB for a 2-fidelity problem initialised with 5 random points at the first fidelity. In the top figures, the solid lines in brown and blue are $f^{(1)}, f^{(2)}$ respectively, and the dashed lines are $\varphi_t^{(1)}, \varphi_t^{(2)}$. The solid green line is $\varphi_t = \min(\varphi_t^{(1)}, \varphi_t^{(2)})$. The small crosses are queries from 1 to $t-1$ and the red star is the maximiser of $\varphi_t$, i.e. the next query $\mathbf{x}_t$. $x_\star$, the optimum of $f^{(2)}$ is shown in magenta. In the bottom figures, the solid orange line is $\beta_t^{1/2}\sigma_{t-1}^{(1)}$ and the dashed black line is $\gamma^{(1)}$. When $\beta_t^{1/2}\sigma_{t-1}^{(1)}(\mathbf{x}_t) \leq \gamma^{(1)}$ we play at fidelity $\mathbf{m}_t = 2$ and otherwise at $\mathbf{m}_t = 1$.

At the initial stages, MF-GP-UCB is mostly exploring $\mathcal{X}$ in the first fidelity. $\beta_t^{1/2}\sigma_{t-1}^{(1)}$ is large and we are yet to constrain $f^{(1)}$ well to proceed to $m = 2$. At $t = 10$, we have constrainted $f^{(1)}$ sufficiently well at a region around the optimum. $\beta_t^{1/2}\sigma_{t-1}^{(1)}(\mathbf{x}_t)$ falls below $\gamma^{(1)}$ and we query at $\mathbf{m}_t = 2$. Notice that once we do this (at $t = 11$), $\varphi_t^{(2)}$ dips to change $\varphi_t$ in that region. At $t = 14$, MF-GP-UCB has identified the maximum $x_\star$ with just 4 queries to $f^{(2)}$. In the last figure, at $t = 50$, the algorithm decides to explore at a point far away from the optimum. However, this query occurs in the first fidelity since we have not sufficiently constrained $f^{(1)}(\mathbf{x}_t)$ in this region. The key idea is that it is *not necessary* to query such regions at the second fidelity as the first fidelity alone is enough to conclude that it is suboptimal. Herein lies the crux of our method. The region shaded in cyan in the last figure is the good set $\mathcal{X}_g = \{x; f^{(2)}(x_\star) - f^{(1)}(x) \leq \zeta^{(1)}\}$ discussed in Section 4. Our analysis predicts that most second fidelity queries in MF-GP-UCB will be confined to this set with high probability and the simulation corroborates this claim. In addition, observe that in a large portion of $\mathcal{X}$, $\varphi_t$ is given by $\varphi_t^{(1)}$ except in a small neighborhood around $x_\star$, where it is given by $\varphi_t^{(2)}$.

the approximations provided via the lower fidelity evaluations. Denoting $\overline{\mathcal{Z}} = \mathcal{X} \backslash \mathcal{Z}$, we decompose $\tilde{R}_n$ as follows,

$$\tilde{R}_n \leq \underbrace{2B \sum_{m=1}^{M-1} \lambda^{(m)} T_n^{(m)}(\mathcal{X})}_{\tilde{R}_{n,1}} + \underbrace{\lambda^{(M)} \sum_{\substack{t:\mathbf{m}_t=M \\ \mathbf{x}_t \in \mathcal{Z}}} \left( f_\star - f^{(M)}(\mathbf{x}_t) \right)}_{\tilde{R}_{n,2}} + \underbrace{\lambda^{(M)} \sum_{\substack{t:\mathbf{m}_t=M \\ \mathbf{x}_t \in \overline{\mathcal{Z}}}} \left( f_\star - f^{(M)}(\mathbf{x}_t) \right)}_{\tilde{R}_{n,3}}.$$

(4)

$\tilde{R}_{n,1}$ is the capital spent on the lower fidelity queries for which we receive no reward. $\tilde{R}_{n,2}$ is the regret due to fidelity $M$ queries in $\mathcal{Z}$ and $\tilde{R}_{n,3}$ is due to fidelity $M$ queries outside $\mathcal{Z}$. To control $\tilde{R}_{n,1}$ we will first bound $T_n^{(m)}(\mathcal{X})$ for $m < M$. This will typically be small containing only polylog$(n)$/poly$(\gamma)$ and $o(n)$ terms. The last two terms can be controlled using the MIGs $\Psi_n$ of $\mathcal{Z}, \overline{\mathcal{Z}}$ respectively (Definition 2). As we will see, $\tilde{R}_{n,2}$ will be the dominant term in $n$ in our final expression since most of the fidelity $M$ queries will be confined to $\mathcal{Z}$. $T_M^{(n)}(\overline{\mathcal{Z}})$ will be sublinear in $n$ and hence $\tilde{R}_{n,3}$ will be of low order. When the lower fidelities allow us to eliminate a large region of the space, $\text{vol}(\mathcal{Z}) \ll \text{vol}(\overline{\mathcal{Z}})$ and consequently the maximum information gain of $\mathcal{Z}$ will be much smaller than that of $\overline{\mathcal{Z}}$, $\Psi_n(\mathcal{Z}) \ll \Psi_n(\overline{\mathcal{Z}})$. As we will see, this results in much better regret for MF-GP-UCB in comparison to GP-UCB.

For the analysis, we will need the following regularity conditions on the kernel. It is satisfied for four times differentiable kernels such as the SE and Matérn kernels with smoothness parameter $\nu > 2$ [10].

**Assumption 8.** *Let $f \sim \mathcal{GP}(\mathbf{0}, \kappa)$, where $\kappa : [0,r]^d \times [0,r]^d \to \mathbb{R}$ is a stationary kernel. The partial derivatives of $f$ satisfies the following high probability bound. There exists constants $a, b > 0$ such that, for all $J > 0$,*

$$\forall\, i \in \{1, \ldots, d\}, \quad \mathbb{P} \left( \sup_x \left| \frac{\partial f(x)}{\partial x_i} \right| > J \right) \leq a e^{-(J/b)^2}.$$

For our proofs we will need to control the conditional variances for queries within a subset $A \subset \mathcal{X}$. To that end, we provide the lemma below.

**Lemma 9.** *Let $f \sim \mathcal{GP}(0, \kappa)$, $f : \mathcal{X} \to \mathbb{R}$ and each time we query at any $x \in \mathcal{X}$ we observe $y = f(x) + \epsilon$, where $\epsilon \sim \mathcal{N}(0, \eta^2)$. Let $A \subset \mathcal{X}$. Assume that we have queried $f$ at $n$ points, $(x_t)_{t=1}^n$ of which $s$ points are in $A$. Let $\sigma_{t-1}^2$ denote the posterior variance at time $t$, i.e. after $t-1$ queries. Then, $\sum_{x_t \in A} \sigma_{t-1}^2(x_t) \leq \frac{2}{\log(1+\eta^{-2})} \Psi_s(A)$.*

*Proof* Let $A_s = \{z_1, z_2, \ldots, z_s\}$ be the queries inside $A$ in the order they were queried. Now, assuming that we have only queried inside $A$ at $A_s$, denote by $\tilde{\sigma}_{t-1}(\cdot)$, the posterior standard deviation after $t-1$ such queries. Then,

$$\sum_{t:x_t \in A} \sigma_{t-1}^2(x_t) \leq \sum_{t=1}^s \tilde{\sigma}_{t-1}^2(z_t) \leq \sum_{t=1}^s \eta^2 \frac{\tilde{\sigma}_{t-1}^2(z_t)}{\eta^2} \leq \sum_{t=1}^s \frac{\log(1 + \eta^{-2}\tilde{\sigma}_{t-1}^2(z_t))}{\log(1 + \eta^{-2})}$$

$$\leq \frac{2}{\log(1 + \eta^{-2})} I(y_{A_s}; f_{A_s})$$

Queries outside $A$ will only decrease the variance of the GP so we can upper bound the first sum by the posterior variances of the GP with only the queries in $A$. The third step uses the inequality $u^2/v^2 \leq \log(1+u^2)/\log(1+v^2)$ with $u = \tilde{\sigma}_{t-1}(z_t)/\eta$ and $v = 1/\eta$ and the last step uses Lemma 6. The result follows from the fact that $\Psi_s(A)$ maximises the mutual information among all subsets of size $s$. ∎

We now proceed to the analysis. To avoid clutter in the notation we will use $\gamma = \gamma^{(m)}$ for all $m$. Generalising this to different $\gamma^{(m)}$'s is straightforward.

Denote $\Delta^{(m)}(x) = f_\star - f^{(m)}(x) - \zeta^{(m)}$ and $\mathcal{J}_\eta^{(m)} = \{x \in \mathcal{X}; \Delta^{(m)}(x) \leq \eta\}$. Let $\tau > 0, \rho > 1$ be given. Central to our analysis will be two partitionings $(\mathcal{X}^{(m)})_{m=1}^M$ and $(\mathcal{H}_\tau^{(m)})_{m=1}^M$ of $\mathcal{X}$. The latter depends on the parameter $\gamma$ and the given $\tau, \rho$. Let $\mathcal{X}^{(1)} = \overline{\mathcal{J}}_0^{(1)}$, $\mathcal{H}_\tau^{(1)} = \overline{\mathcal{J}}_{\max(\tau,\rho\gamma)}^{(1)}$. Then define,

$$\mathcal{X}^{(m)} = \overline{\mathcal{J}}_0^{(m)} \cap \left( \bigcap_{\ell=1}^{m-1} \mathcal{J}_0^{(\ell)} \right) \quad \text{for } 2 \leq m \leq M-1, \qquad \mathcal{X}^{(M)} = \bigcap_{\ell=1}^{M-1} \mathcal{J}_0^{(\ell)}. \qquad (5)$$

$$\mathcal{H}_\tau^{(m)} = \overline{\mathcal{J}}_{\max(\tau,\rho\gamma)}^{(m)} \cap \left( \bigcap_{\ell=1}^{m-1} \mathcal{J}_{\max(\tau,\rho\gamma)}^{(\ell)} \right) \quad \text{for } 2 \leq m \leq M-1, \qquad \mathcal{H}_\tau^{(M)} = \bigcap_{\ell=1}^{M-1} \mathcal{J}_{\max(\tau,\rho\gamma)}^{(\ell)}.$$

In addition to the above, we will also find it useful to define the sets "above" $\mathcal{H}_\tau^{(m)}$ as $\widehat{\mathcal{H}}_\tau^{(m)} = \bigcup_{\ell=m+1}^M \mathcal{H}_\tau^{(\ell)}$ and the sets "below" $\mathcal{H}_\tau^{(m)}$ as $\widetilde{\mathcal{H}}_\tau^{(m)} = \bigcup_{\ell=1}^{m-1} \mathcal{H}_\tau^{(\ell)}$. Intuitively, $\mathcal{H}_\tau^{(m)}$ is the set of points that MF-GP-UCB will query at the $m^{\text{th}}$ fidelity but exclude from higher fidelities due to information from fidelity $m$. $\widetilde{\mathcal{H}}_\tau^{(m)}$ is the set of points that can be excluded from queries at fidelities $m$ and beyond due to information from lower fidelities. $\widehat{\mathcal{H}}_\tau^{(m)}$ are points that need to be queried at fidelities higher than $m$. In the 2 fidelity setting described in Section 4, the set $\mathcal{X}_g$ is $\mathcal{X}^{(2)}$ and $\widetilde{\mathcal{X}}_{g,\rho}$ is $\mathcal{H}^{(2)}$. Finally, for any given $\alpha > 0$ we will also define $\mathcal{H}_{\tau,n}^{(m)} = \{x \in \mathcal{X} : \text{B}_2(x, r\sqrt{d}/n^{\frac{\alpha}{2d}}) \cap \mathcal{H}_\tau^{(m)} \neq \varnothing \wedge x \notin \widehat{\mathcal{H}}^{(m)}\}$ to be an $n$-dependence inflation of $\mathcal{H}_{\tau,n}^{(m)}$. Here, $\text{B}_2(x,\epsilon)$ is an $L_2$ ball of radius $\epsilon$ centred at $x$. The sets $\{\mathcal{H}_{\tau,n}^{(m)}\}_{m=1}^M$ depend on $\rho, \gamma, \tau, n$ and $\alpha$. Notice that for any $\alpha > 0$, as $n \to \infty$, $\mathcal{H}_{\tau,n}^{(m)} \to \mathcal{H}_\tau^{(m)}$. In addition to the above, denote the $\varepsilon$ covering number of a set $A \subset \mathcal{X}$ in the $\|\cdot\|_2$ metric by $\Omega_\varepsilon(A)$. Let $T_n^{(m)}(A)$ denote the number of queries in a subset $A \subset \mathcal{X}$ at fidelity $m$. $\mathcal{D}_n^{(m)} = \{(\mathbf{x}_t, \mathbf{y}_t)\}_{t:\mathbf{m}_t=m}$ denotes the set of query-value pairs at the $m^{\text{th}}$ fidelity until time $n$. Our main theorem is as follows.

**Theorem 10.** *Let $\mathcal{X} \subset [0,r]^d$ be compact and convex. Let $f^{(m)} \sim \mathcal{GP}(\mathbf{0}, \kappa) \; \forall m$, and satisfy assumptions A2, A3. Let $\kappa$ satisfy Assumption 8 with some constants $a, b$. Pick $\delta \in (0,1)$ and run* MF-GP-UCB *with*

$$\beta_t = 2\log\left(\frac{M\pi^2 t^2}{2\delta}\right) + 4d\log(t) + \max\left\{0, 2d\log\left(brd\log\left(\frac{6Mad}{\delta}\right)\right)\right\}.$$

*For all $\alpha \in (0,1), \tau > 0, \rho > \rho_0 = \max\{2, 1 + \sqrt{(1+2/\alpha)/(1+d)}\}$ and sufficiently large $\Lambda$, we have $R(\Lambda) \in \mathcal{O}\left(\sum_{m=1}^M \lambda^{(m)}\sqrt{n_\Lambda \beta_{n_\Lambda} \Psi_{n_\Lambda}(\mathcal{H}_{\tau,n_\Lambda}^{(m)})} + \frac{\text{diam}(\widehat{\mathcal{H}}_\tau^{(m)})^d \text{polylog}(n_\Lambda)}{\text{poly}(\gamma)}\right)$. Here, $n_\Lambda = \lfloor \Lambda/\lambda^{(M)} \rfloor$ as before.*
*Precisely, there exists $\Lambda_0$ such that for all $\Lambda \geq \Lambda_0$, with probability $> 1 - \delta$ we have,*

$$R(\Lambda) \leq 2B\lambda^{(M)} + \lambda^{(M)}\left[\sqrt{2C_1 M n_\Lambda^\alpha \Psi_{2Mn_\Lambda^\alpha}(\widetilde{\mathcal{H}}^{(M)})} + \sqrt{2C_1 n_\Lambda \Psi_{2n_\Lambda}(\mathcal{H}_{\tau,n_\Lambda}^{(M)})} + \frac{\pi^2}{6}\right]$$

$$+ 2B\sum_{m=1}^{M-1}\lambda^{(m)}\left[(m-1)(2n_\Lambda^\alpha) + \frac{1}{\tau}\left(\sqrt{2C_1 n_\Lambda \beta_{2n_\Lambda}\Psi_{2n_\Lambda}(\mathcal{H}_{\tau,n_\Lambda}^{(m)})} + \frac{\pi^2}{6}\right) + \right.$$

$$\left. \Omega_{\varepsilon_n}(\widehat{\mathcal{H}}_\tau^{(m)})\left(\frac{2\eta^2}{\gamma^2}\beta_n + 1\right)\right],$$

*where $C_1 = 8/\log(1+\eta^2)$. For the SE kernel $\varepsilon_n = \frac{\gamma}{\sqrt{8C_{SE}\beta_n}}$, and therefore $\Omega_{\varepsilon_n}(\widehat{\mathcal{H}}^{(m)}) \in \mathcal{O}\left(\frac{\text{diam}(\widehat{\mathcal{H}}^{(m)})^d(\log(n))^{d/2}}{\gamma^d}\right)$. For the Matérn kernel $\varepsilon_n = \frac{\gamma^2}{8C_{Mat}\beta_n}$ and therefore $\Omega_{\varepsilon_n}(\widehat{\mathcal{H}}^{(m)}) \in \mathcal{O}\left(\frac{\text{diam}(\widehat{\mathcal{H}}^{(m)})^d(\log(n))^d}{\gamma^{2d}}\right)$. $C_{SE}, C_{Mat}$ are kernel dependent constants. As $\Lambda \to \infty$, $n_\Lambda \to \infty$ and hence $\mathcal{H}_{\tau,n_\Lambda}^{(m)} \to \mathcal{H}_\tau^{(m)}$ for all $m \in \{1, \ldots, M\}$ and $\alpha \in (0,1)$.*

**Synopsis:** Ignoring the common terms, constants and $n_\Lambda^\alpha$ terms, the regret for GP-UCB is $\lambda^{(M)}\sqrt{n_\Lambda \Psi_{n_\Lambda}(\mathcal{X})}$ whereas for MF-GP-UCB it is $\sum_m \lambda^{(m)}\sqrt{n_\Lambda \Psi_{n_\Lambda}(\mathcal{H}_{\tau,n}^{(m)})}$. In problems where

Figure 7: Illustration of the sets $\{\mathcal{F}_n^{(\ell)}\}_{\ell=1}^{m-1}$ with respect to $\mathcal{H}_\tau^{(m)}$. The grid represents a $r\sqrt{d}/n^{\alpha/2d}$ covering of $\mathcal{X}$. The yellow region is $\widehat{\mathcal{H}}_\tau^{(m)}$. The area enclosed by the solid red line (excluding $\widehat{\mathcal{H}}_\tau^{(m)}$) is $\mathcal{H}_\tau^{(m)}$. $\mathcal{H}_{\tau,n}^{(m)}$, shown by a dashed red line, is obtained by inflating $\mathcal{H}_\tau^{(m)}$ by $r\sqrt{d}/n^{\alpha/2d}$. The grey shaded region represents $\bigcup_{\ell=1}^{m-1}\mathcal{F}_n^{(\ell)}$. By our definition, $\bigcup_{\ell=1}^{m-1}\mathcal{F}_n^{(\ell)}$ contains the cells which are entirely outside $\mathcal{H}_\tau^{(m)}$. However, the inflation $\mathcal{H}_{\tau,n}^{(m)}$ is such that $\widehat{\mathcal{H}}_\tau^{(m)} \cup \mathcal{H}_{\tau,n}^{(m)} \cup \bigcup_{\ell=1}^{m-1}\mathcal{F}_n^{(\ell)} = \mathcal{X}$. As $n \to \infty$, $\mathcal{H}_{\tau,n}^{(m)} \to \mathcal{H}_\tau^{(m)}$.

$\text{vol}(\mathcal{H}_{\tau,n}^{(m)}) \ll \text{vol}(\mathcal{H}_{\tau,n}^{(m)})$, and $\lambda^{(m)} \ll \lambda^{(m+1)}$ MF-GP-UCB achieves signficantly better regret than GP-UCB. When the sets become larger (the approximation becomes worse) and the costs become comparable the bound decays gracefully. The $\lambda^{(m)}\sqrt{n_\Lambda^\alpha \Psi_{n_\Lambda^\alpha}(\mathcal{H}_{\tau,n}^{(m)})}$ terms can be made arbitrarily small by picking large enough $\rho$, provided $\mathcal{H}_{\tau,n}^{(m)}$ is still small relative to $\mathcal{X}$. On the other hand the $\text{diam}(\widehat{\mathcal{H}}_\tau^{(m)})\text{polylog}(n_\Lambda)/\text{poly}(\gamma)$ terms could be big if $\gamma$ is too small. MF-GP-UCB requires that $\gamma$ will be chosen large enough so that the above term remains small relative to $\sqrt{n_\Lambda \beta_{n_\Lambda} \Psi_{n_\Lambda}(\mathcal{H}_\tau^{(m)})}$ which is not too restrictive since we expect $\widehat{\mathcal{H}}_\tau^{(m)}$ to be much smaller than $\mathcal{H}_\tau^{(m)}$. Our analysis reveals that an optimal choice for the SE kernel scales $\gamma^{(m)} \asymp (\lambda^{(m)}\zeta^{(m)}/(t\lambda^{(m+1)}))^{1/(d+2)}$ at time step $t$. However this observation is of little practical consequence as the leading constant depends on several problem dependent quantities such as $\Psi_n(\mathcal{X}_g)$. Our heuristics for setting $\gamma$ seemed to work well in practice (see Section 5).

***Proof of Theorem 10.*** We will study MF-GP-UCB after $n$ time steps regardless of the queried fidelities and bound $\tilde{R}_n$. Then we will bound the number of plays $N$ within capital $\Lambda$. For the analysis, at time $n$ we will consider a $\frac{r\sqrt{d}}{2n^{\frac{\alpha}{2d}}}$-covering of the space $\mathcal{X}$ of size $n^{\frac{\alpha}{2}}$. For instance, if $\mathcal{X} = [0, r]^d$ a sufficient discretisation would be an equally spaced grid having $n^{\alpha/2d}$ points per side. Let $\{a_{i,n}\}_{i=1}^{n^{\frac{\alpha}{2}}}$ be the points in the covering, $F_n = \{A_{i,n}\}_{i=1}^{n^{\frac{\alpha}{2}}}$ be the cells in the covering, i.e. $A_{i,n}$ is the set of points which are closest to $a_{i,n}$ in the covering. Next we define another partitioning of the space similar in spirit to (5) using this partitioning. First let $F_n^{(1)} = \{A_{i,n} \in F_n : A_{i,n} \subset \mathcal{J}_{\max(\tau,\rho\gamma)}^{(1)}\}$. Next,

$$F_n^{(m)} = \left\{ A_{i,n} \in F_n : A_{i,n} \subset \overline{\mathcal{J}}_{\max(\tau,\rho\gamma)}^{(m)} \quad \wedge \quad A_{i,n} \notin \bigcup_{\ell=1}^{m-1} F_n^{(\ell)} \right\} \quad \text{for } 2 \leq m \leq M-1. \tag{6}$$

Note that $F_n^{(m)} \subset F_n$. We define the following *disjoint* subsets $\{\mathcal{F}_n^{(m)}\}_{m=1}^{M-1}$ of $\mathcal{X}$ via $\mathcal{F}_n^{(m)} = \bigcup_{A_{i,n} \in F_n^{(m)}} A_{i,n}$. We have illustrated $\bigcup_{\ell=1}^{m-1}\mathcal{F}_n^{(\ell)}$ with respect to $\mathcal{H}_\tau^{(m)}$ in Figure 7. By noting that $\mathcal{H}_{\tau,n}^{(1)} = \mathcal{H}^{(1)}$ we make the following observation,

$$T_n^{(m)}(\mathcal{X}) \leq \sum_{\ell=1}^{m-1} T_n^{(m)}(\mathcal{F}_n^{(\ell)}) + T_n^{(m)}(\mathcal{H}_{\tau,n}^{(m)}) + T_n^{(m)}(\widehat{\mathcal{H}}^{(m)}). \tag{7}$$

This follows by noting that $\overline{\mathcal{H}_{\tau,n}^{(m)} \cup \widehat{\mathcal{H}}^{(m)}} \subset \bigcup_{\ell=1}^{m-1}\mathcal{F}_n^{(\ell)}$ (See Fig. 7). To control $\tilde{R}_n$ we will bound control each of these terms individually. First we focus on $\widehat{\mathcal{H}}^{(m)}$ for which we use the following lemma. The proof is given in Section C.0.1.

**Lemma 11.** *Let $f \sim \mathcal{GP}(\mathbf{0}, \kappa)$, $f : \mathcal{X} \to \mathbb{R}$ and we observe $y = f(x) + \epsilon$ where $\epsilon \sim \mathcal{N}(0, \eta^2)$. Let $A \subset \mathcal{X}$ such that its $L_2$ diameter $\text{diam}(A) \leq D$. Say we have $n$ queries $(\mathbf{x}_t)_{t=1}^n$ of which $s$ points*

*are in A. Then the posterior variance of the GP, $\kappa'(x,x)$ at any $x \in A$ satisfies*

$$\kappa'(x,x) \leq \begin{cases} C_{SE}D^2 + \frac{\eta^2}{s} & \text{if } \kappa \text{ is the SE kernel,} \\ C_{Mat}D + \frac{\eta^2}{s} & \text{if } \kappa \text{ is the Matérn kernel,} \end{cases}$$

*for appropriate constants $C_{SE}, C_{Mat}$.*

First consider the SE kernel. At time $t$ consider any $\varepsilon_n = \frac{\gamma}{\sqrt{8C_{SE}\beta_n}}$ covering $(B_i)_{i=1}^{\varepsilon_n}$ of $\widehat{\mathcal{H}}^{(m)}$. The number of queries inside any $B_i$ of this covering at time $n$ will be at most $\frac{2\eta^2}{\gamma^2}\beta_n + 1$. To see this, assume we have already queried $2\eta^2/\gamma^2 + 1$ times inside $B_i$ at time $t \leq n$. By Lemma 11 the maximum variance in $A_i$ can be bounded by

$$\max_{x \in A_i} \kappa_{t-1}^{(m)}(x,x) \leq C_{SE}(2\varepsilon_n)^2 + \frac{\eta^2}{T_t^{(m)}(A_i)} < \frac{\gamma^2}{\beta_n}.$$

Therefore, $\beta_t^{1/2}\sigma_{t-1}^{(m)}(x) \leq \beta_n^{1/2}\sigma_{t-1}^{(m)}(x) < \gamma$ and we will not query inside $A_i$ until time $n$. Therefore, the number of $m^{\text{th}}$ fidelity queries is bounded by $\Omega_{\varepsilon_n}(\widehat{\mathcal{H}}^{(m)})\left(\frac{2\eta^2}{\gamma^2}\beta_n + 1\right)$. The proof for the Matérn kernel follows similarly using $\varepsilon_n = \frac{\gamma^2}{8C_{Mat}\beta_n}$. Next, we bound $T_n^{(m)}(\mathcal{H}_{\tau,n}^{(m)})$ for which we will use the following Lemma. The proof is given in Section C.0.2.

**Lemma 12.** *For $\beta_t$ as given in Theorem 10, we have the following with probability $> 1 - 5\delta/6$.*

$$\forall m \in \{1, \dots, M\}, \ \forall t \geq 1, \quad \Delta^{(m)}(\mathbf{x}_t) = f_\star - f^{(m)}(\mathbf{x}_t) \leq 2\beta_t \sigma_{t-1}^{(m)}(\mathbf{x}_t) + 1/t^2.$$

First, we will analyse the quantity $\tilde{R}_n^{(m)} = \sum_{\substack{t:\mathbf{m}_t=m \\ \mathbf{x}_t \in \mathcal{H}_{\tau,n}^{(m)}}} \Delta^{(m)}(\mathbf{x}_t)$ for $m < M$. Lemma 12 gives us $\tilde{R}_n^{(m)} \leq 2\beta_n^{1/2}\sum \sigma_{t-1}^{(m)}(\mathbf{x}_t) + \pi^2/6$. Then, using Lemma 9 and Jensen's inequality we have,

$$\left(\tilde{R}_n^{(m)} - \frac{\pi^2}{6}\right)^2 \leq 4\beta_t \, T_n^{(m)}(\mathcal{H}_{\tau,n}^{(m)}) \sum_{\substack{t:\mathbf{m}_t=m \\ \mathbf{x}_t \in \mathcal{H}_{\tau,n}^{(m)}}} \left(\sigma_{t-1}^{(m)}\right)^2(\mathbf{x}_t) \leq C_1\beta_t \, T_n^{(m)}(\mathcal{H}_{\tau,n}^{(m)})\Psi_{T_n^{(m)}(\mathcal{H}_{\tau,n}^{(m)})}(\mathcal{H}_{\tau,n}^{(m)}).$$

$$(8)$$

We therefore have, $\tilde{R}_n^{(m)} \leq \sqrt{C_1 n\beta_n \Psi_n(\mathcal{H}_{\tau,n}^{(m)})} + \pi^2/6$ since trivially $T_n^{(m)}(\mathcal{H}_{\tau,n}^{(m)}) < n$. However, since $\Delta^{(m)}(x) > \tau$ for $x \in \mathcal{H}_{\tau,n}^{(m)}$ we have $T_n^{(m)}(\mathcal{H}_{\tau,n}^{(m)}) < \frac{1}{\tau}\left(\sqrt{C_1 n\beta_n \Psi_n(\mathcal{H}_{\tau,n}^{(m)})} + \pi^2/6\right)$.

**Remark 13.** *Since $\Psi_n(\cdot)$ is typically sublinear in $n$, it is natural to ask if we can recursively apply this to obtain a tighter bound on $T_n^{(m)}(\mathcal{H}_{\tau,n}^{(m)})$. For instance, since $\Psi_n(\cdot)$ is $\text{polylog}(n)$ for the SE kernel (Srinivas et al. [28], Theorem 5) by repeating the argument above once we get, $T_n^{(m)}(\mathcal{H}_{\tau,n}^{(m)}) \in \mathcal{O}\left(\frac{1}{\tau^{3/2}}\sqrt{C_1 n^{1/2}\text{polylog}(n)\beta_n\Psi_{\tau^{-3/2}n^{1/2}\text{polylog}(n)}(\mathcal{H}_{\tau,n}^{(m)})}\right)$. However, while this improves the dependence on $n$ it worsens the dependence on $\tau$. In fact, using a discretisation argument similar to that in Lemma 14 and the variance bound in Lemma 11, a $\text{polylog}(n)/\text{poly}(\tau)$ bound can be shown, with the $\text{poly}(\tau)$ term being $\tau^{d+2}$ for the SE kernel and $\tau^{2d+2}$ for the Matérn kernel. In fact, the same argument can be applied to GP-UCB to show that the number of plays on a $\tau$-suboptimal set is $\text{polylog}(n)/\text{poly}(\tau)$. If we are to avoid this $1/\text{poly}(\tau)$ dependence for GP-UCB the best you can achieve for GP-UCB is a $\mathcal{O}(n^{1/2})$ rate for the SE kernel and $\mathcal{O}(n^{\frac{1}{2}+\frac{d(d+1)}{2\nu+d(d+1)}})$ for the Matérn kernel.*

Finally, to control the first term in (7), we will bound $T_n^{(>m)}(\mathcal{F}_n^{(m)})$. To that end we provide the following Lemma. The proof is given in Section C.0.3.

**Lemma 14.** *Consider any $A_{i,n} \in F_n^{(m)}$ where $F_n^{(m)}$ is as defined in (6) for any $\alpha \in (0,1)$. Let $\rho, \beta_t$ be as given in Theorem 10, Then for all $u \geq \max\{3, (2(\rho - \rho_0)\eta)^{-2/3}\}$ we have,*

$$\mathbb{P}(T_n^{(>m)}(A_{i,n}) > u) \leq \frac{\delta}{\pi^2} \cdot \frac{1}{u^{1+4/\alpha}}$$

We will use the above result with $u = n^{\alpha/2}$. Applying the union bound we have,

$$\mathbb{P}\left(\forall\, m \in \{1, \ldots, M\},\ T_n^{(>m)}(\mathcal{F}_n^{(m)}) > |F_n^{(m)}| n^{\alpha/2}\right) \leq \sum_{m=1}^{M} \mathbb{P}\left(T_n^{(>m)}(\mathcal{F}_n^{(m)}) > |F_n^{(m)}| n^{\alpha/2}\right)$$

$$\leq \sum_{m=1}^{M} \sum_{A_{i,n} \in F_n^{(m)}} \mathbb{P}\left(T_n^{(>m)}(A_{i,n}) > n^{\alpha/2}\right) \leq \sum_{m=1}^{M} |F_n^{(m)}| \frac{\delta}{\pi^2} \frac{1}{n^{2+\alpha/2}} \leq |F_n| \frac{\delta}{\pi^2} \frac{1}{n^{2+\alpha/2}} = \frac{\delta}{\pi^2} \frac{1}{n^2}$$

Applying the union bound once again, we have $T_n^{(>m)}(\mathcal{F}_n^{(m)}) \leq n^\alpha$ for all $m$ and all $n \geq \max\{3, (2(\rho - \rho_0)\eta)^{2/3}\}^{2/\alpha}$ with probability $> 1 - \delta/6$. Henceforth, all statements we make will make use of the results in Lemmas 11, 12 and 14 and will hold with probability $> 1 - \delta$.

First using equation (7) and noting $T_n^{(m)}(\mathcal{F}_n^{(\ell)}) \leq T_n^{(>\ell)}(\mathcal{F}_n^{(\ell)})$ for $\ell < m$ we bound $T_n^{(m)}(\mathcal{X})$ for $m < M$.

$$T_n^{(m)}(\mathcal{X}) \leq (m-1)n^\alpha + \frac{1}{\tau}\left(\sqrt{C_1 n \beta_n \Psi_n(\mathcal{H}_{\tau,n}^{(m)})} + \frac{\pi^2}{6}\right) + \Omega_{\varepsilon_n}(\widehat{\mathcal{H}}^{(m)})\left(\frac{2\eta^2}{\gamma^2}\beta_n + 1\right).$$

Using this bound we can control $\tilde{R}_{n,1}$ in (4). To bound $\tilde{R}_{n,2}$ and $\tilde{R}_{n,3}$ we set $\mathcal{Z} = \mathcal{H}_{\tau,n_\Lambda}^{(M)}$ and use Lemma 12 noting that when $\mathbf{m}_t = M$, $r_t = \Delta^{(M)}(\mathbf{x}_t)$. Using similar calculations to (8) and as $T_n^{(M)}(\mathcal{H}_{\tau,n}^{(m)}) \leq n$, we have $\tilde{R}_{n,2} \leq \sqrt{C_1 n \beta_n \Psi_n(\mathcal{H}_{\tau,n}^{(m)})} + \sum_{\mathbf{x}_t \in \mathcal{Z}} 1/t^2$. Next, using Lemma 14 and observing $\overline{\mathcal{Z}} = \overline{\mathcal{H}_{\tau,n}^{(M)}} \subset \bigcup_{\ell=1}^{M-1} \mathcal{F}_n^{(m)} \subset \breve{\mathcal{H}}^{(M)}$, we have,

$$\tilde{R}_{n,3} = \sum_{\substack{t:\mathbf{m}_t=M \\ \mathbf{x}_t \in \overline{\mathcal{Z}}}} \left(f_\star - f^{(M)}(\mathbf{x}_t)\right) \leq \sum_{\substack{t:\mathbf{m}_t=M \\ \mathbf{x}_t \in \bigcup_{\ell=1}^{M-1} \mathcal{F}_n^{(m)}}} 2\beta_t^{1/2} \sigma_{t-1}^{(m)}(\mathbf{x}_t) + \sum_{\mathbf{x}_t \in \overline{\mathcal{Z}}} \frac{1}{t^2}$$

$$\leq \sqrt{C_1 M n^\alpha \beta_n \Psi_{Mn^\alpha}(\breve{\mathcal{H}}^{(M)})} + \sum_{\mathbf{x}_t \in \overline{\mathcal{Z}}} \frac{1}{t^2}.$$

Plugging these bounds back into (4), we obtain a bound on the regret similar to the one given in the theorem except with $n$ replaced by $2n_\Lambda$. The last step in the proof will be to show that for sufficiently large $\Lambda$, $N \leq 2n_\Lambda$ which will complete the proof. For this we turn back to our bounds for $T_n^{(m)}(\mathcal{X})$, $m < M$. Next, we can show that the following term upper bounds the number of queries at fidelities less than $M$,

$$(M-1)n^\alpha + \sum_{m=1}^{M-1} \frac{1}{\tau}\left(\sqrt{2C_1 n_\Lambda \beta_{2n_\Lambda} \Psi_{2n_\Lambda}(\mathcal{H}_{\tau,n_\Lambda}^{(m)})} + \frac{\pi^2}{6}\right) + \sum_{m=1}^{M-1} \Omega_{\varepsilon_n}(\widehat{\mathcal{H}}^{(m)})\left(\frac{2\eta^2}{\gamma^2}\beta_n + 1\right).$$

Assume $n_0$ is large enough so that $n_0 \geq \max\{3, (2(\rho - \rho_0)\eta)^{-2/3}\}^{2/\alpha}$ and for all $n \geq n_0$, $n/2$ is larger than the above upper bound. We can find such an $n_0$ since the bound is $o(n)$. Therefore, for all $n \geq n_0$, $T_n^{(M)}(\mathcal{X}) > n/2$. Since our bounds hold with probability $> 1 - \delta$ uniformly over $n$ we can invert the above inequality to bound the number of plays $N$ after capital $\Lambda$: $N \leq 2\Lambda/\lambda^{(M)}$ with probability $> 1 - \delta$ if $\Lambda \geq \Lambda_0 = \lambda^{(M)}(n_0 + 1)$. The theorem follows with the observation $N \geq n_\Lambda \implies \mathcal{H}_{\tau,N}^{(m)} \subset \mathcal{H}_{\tau,n_\Lambda}^{(m)} \implies \Psi_N(\mathcal{H}_{\tau,N}^{(m)}) \leq \Psi_N(\mathcal{H}_{\tau,n_\Lambda}^{(m)}) \leq \Psi_{2n_\Lambda}(\mathcal{H}_{\tau,n_\Lambda}^{(m)})$. ∎

### C.0.1 Proof of Lemma 11

Since the posterior variance only decreases with more observations, we can upper bound $\kappa'(x,x)$ for any $x \in A$ by considering its posterior variance with only the $s$ observations in $A$. Next the maximum variance within $A$ occurs if we pick 2 points $x_1, x_2$ that are distance $D$ apart and have all observations at $x_1$; then $x_2$ has the highest posterior variance. Therefore, we will bound $\kappa'(x,x)$ for any $x \in A$ with $\kappa(x_2, x_2)$ in the above scenario. Let $\kappa_0 = \kappa(x,x)$ and $\kappa(x, x') = \kappa_0 \phi(\|x - x'\|_2)$, where $\phi(\cdot) \leq 1$ depends on the kernel. Denote the gram matrix in the scenario described above by $\Delta = \kappa_0 \mathbf{1}\mathbf{1}^\top + \eta^2 I$. Then using the Sherman-Morrison formula, the posterior variance (1) can be bounded via,

$$\kappa'(x,x) \leq \kappa'(x_2, x_2) = \kappa(x_2, x_2) - [\kappa(x_1, x_2)\mathbf{1}]^\top \Delta^{-1} [\kappa(x_1, x_2)\mathbf{1}]$$

$$= \kappa_0 - \kappa_0\phi^2(D)\mathbf{1}^\top \left[ \frac{\kappa_0}{\eta^2}I - \frac{\left(\frac{\kappa_0}{\eta^2}\right)^2 \mathbf{1}\mathbf{1}^\top}{1 + \frac{\kappa_0}{\eta^2}s} \right] \mathbf{1} = \kappa_0 - \kappa_0\phi^2(D)\left( \frac{\kappa_0}{\eta^2}s - \frac{\left(\frac{\kappa_0}{\eta^2}\right)^2 s^2}{1 + \frac{\kappa_0}{\eta^2}s} \right)$$

$$= \kappa_0 - \kappa_0\phi^2(D)\frac{s}{\frac{\eta^2}{\kappa_0} + s} = \frac{1}{1 + \frac{\eta^2}{\kappa_0 s}}\left( \kappa_0 - \kappa_0\phi^2(D) + \frac{\eta^2}{s} \right)$$

$$\leq \kappa_0(1 - \phi^2(D)) + \frac{\eta^2}{s}.$$

For the SE kernel $\phi^2(D) = \exp\left(\frac{-D^2}{2h^2}\right)^2 = \exp\left(\frac{-D^2}{h^2}\right) \leq 1 - \frac{D^2}{h^2}$. Plugging this into the bound above retrieves the first result with $C_{SE} = \kappa_0/h^2$. For the Matérn kernel we use a Lipschtiz constant $L_{Mat}$ of $\phi$. Then $1 - \phi^2(D) = (1 - \phi(D))(1 + \phi(D)) \leq 2(\phi(0) - \phi(D)) \leq 2L_{Mat}D$. We get the second result with $C_{Mat} = 2\kappa_0 L_{Mat}$. Since the SE kernel decays fast, we get a stronger result on its posterior variance which translates to a better bound in our theorems. ∎

### C.0.2 Proof of Lemma 12

The first part of the proof mimics the arguments in Lemmas 5.6, 5.7 of Srinivas et al. [28]. By assumption 8 and the union bound we can show,

$$\mathbb{P}\left( \forall m \in \{1,\ldots,M\}, \; \forall i \in \{1,\ldots,d\}, \; \forall x \in \mathcal{X}, \quad \left| \frac{\partial f^{(m)}(x)}{\partial x_i} \right| < b\log\left(\frac{6Mad}{\delta}\right) \right) \geq 1 - \frac{\delta}{6}.$$

Now we construct a discretisation $F_t$ of $\mathcal{X}$ of size $(\nu_t)^d$ such that we have for all $x \in \mathcal{X}$, $\|x - [x]_t\|_1 \leq rd/\nu_t$. Here $[x]_t$ is the closest point to $x$ in the discretisation. (Note that this is different from the discretisation appearing in Theorem 10 even though we have used the same notation). By choosing $\nu_t = t^2 brd\sqrt{6Mad/\delta}$ and using the above we have

$$\forall x \in \mathcal{X}, \quad |f^{(m)}(x) - f^{(m)}([x]_t)| \leq b\log(6Mad/\delta)\|x - [x]_t\|_1 \leq 1/t^2 \tag{9}$$

for all $f^{(m)}$'s with probability $> 1 - \delta/6$.

Noting that $\beta_t \geq 2\log(M|F_t|\pi^2 t^2/2\delta)$ for the given choice of $\nu_t$ we have the following with probability $> 1 - \delta/3$.

$$\forall t \geq 1, \; \forall m \in \{1,\ldots,M\}, \; \forall a \in F_t, \quad |f^{(m)}(a) - \mu_{t-1}^{(m)}(a)| \leq \beta_t^{1/2}\sigma_{t-1}^{(m)}(a). \tag{10}$$

The proof uses Gaussian concentration by only conditioning on $\mathcal{D}_t^{(m)}$. Note that instead of a fixed set over all $t$, we change the set at which we have confidence based on the discretisation. Similarly we can show that with probability $> 1 - \delta/3$ we also have confidence on the decisions $\mathbf{x}_t$ at all time steps. Precisely,

$$\forall t \geq 1, \; \forall m \in \{1,\ldots,M\}, \quad |f^{(m)}(\mathbf{x}_t) - \mu_{t-1}^{(m)}(\mathbf{x}_t)| \leq \beta_t^{1/2}\sigma_{t-1}^{(m)}(\mathbf{x}_t). \tag{11}$$

Using (9),(10) and (11) the following statements hold with probability $> 1 - 5\delta/6$. First, using assumption **A2** we can upper bound $f_\star$ by,

$$f_\star \leq f^{(m)}(x_\star) + \zeta^{(m)} \leq f^{(m)}([x_\star]_t) + \zeta^{(m)} + \frac{1}{t^2} \leq \varphi_t^{(m)}([x_\star]_t) + \frac{1}{t^2}. \tag{12}$$

Since the above holds for all $m$, we have $f_\star \leq \varphi_t([x_\star]_t) + 1/t^2$. Now, we bound $\Delta^{(m)}(\mathbf{x}_t)$.

$$\Delta^{(m)}(\mathbf{x}_t) = f_\star - f^{(m)}(\mathbf{x}_t) - \zeta^{(m)} \leq \varphi_t([x_\star]_t) + \frac{1}{t^2} - f^{(m)}(\mathbf{x}_t) - \zeta^{(m)}$$

$$\leq \varphi_t(\mathbf{x}_t) - f^{(m)}(\mathbf{x}_t) - \zeta^{(m)} + \frac{1}{t^2} \leq \varphi_t^{(m)}(\mathbf{x}_t) - \mu_{t-1}^{(M)}(\mathbf{x}_t) + \beta_t^{1/2}\sigma_{t-1}^{(M)}(\mathbf{x}_t) - \zeta^{(m)} + \frac{1}{t^2}$$

$$\leq 2\beta_t^{1/2}\sigma_{t-1}^{(M)}(\mathbf{x}_t) + \frac{1}{t^2}. \quad ∎$$

### C.0.3 Proof of Lemma 14

First, we will invoke the same discretisation used in the proof of Lemma 12 via which we have $\varphi_t([x_\star]_t) \geq f_\star - 1/t^2$ (12). (Therefore, Lemma 14 holds only with probability $> 1 - \delta/6$, but this event has already been accounted for in Lemma 12.) Let $b_{i,n,t} = \operatorname{argmax}_{x \in A_{i,n}} \varphi_t(x)$ be the maximiser of the upper confidence bound in $A_{i,n}$ at time $t$. Now using the relaxation $\mathbf{x}_t \in A_{i,n} \implies \varphi_t(b_{i,n,t}) > \varphi_t([x_\star]_t) \implies \varphi_t^{(m)}(b_{i,n,t}) > f_\star - 1/t^2$ and proceeding,

$$\mathbb{P}(T_n^{(>m)}(A_{i,n}) > u) \leq \mathbb{P}\big(\exists t : u+1 \leq t \leq n, \ \varphi_t^{(m)}(b_{i,n,t}) > f_\star - 1/t^2 \quad \wedge \quad \beta_t^{1/2}\sigma_{t-1}^{(m)}(b_{i,n,t}) < \gamma\big)$$

$$\leq \sum_{t=u+1}^{n} \mathbb{P}\big(\mu_{t-1}^{(m)}(b_{i,n,t}) - f^{(m)}(b_{i,n,t}) > \Delta^{(m)}(b_{i,n,t}) - \beta_t^{1/2}\sigma_{t-1}^{(m)}(b_{i,n,t}) - 1/t^2 \quad \wedge$$

$$\beta_t^{1/2}\sigma_{t-1}^{(m)}(b_{i,n,t}) < \gamma\big)$$

$$\leq \sum_{t=u+1}^{n} \mathbb{P}\big(\mu_{t-1}^{(m)}(b_{i,n,t}) - f^{(m)}(b_{i,n,t}) > (\rho-1)\beta_t^{1/2}\sigma_{t-1}^{(m)}(b_{i,n,t}) - 1/t^2\big)$$

$$\leq \sum_{t=u+1}^{n} \mathbb{P}_{Z \sim \mathcal{N}(0,1)}\left(Z > (\rho_0 - 1)\beta_t^{1/2}\right) \leq \sum_{t=u+1}^{n} \frac{1}{2}\exp\left(\frac{(\rho_0 - 1)^2}{2}\beta_t\right) \qquad (13)$$

$$\leq \frac{1}{2}\left(\frac{\delta}{M\pi^2}\right)^{(\rho_0-1)^2} \sum_{t=u+1}^{n} t^{-(\rho_0-1)^2(2+2d)} \leq \frac{\delta}{M\pi^2}u^{-(\rho_0-1)^2(2+2d)+1} \leq \frac{\delta}{\pi^2}\frac{1}{u^{1+4/\alpha}}.$$

In the second step we have rearranged the terms and used the definition of $\Delta^{(m)}(x)$. In the third step, as $A_{i,n} \subset \overline{\mathcal{J}}_{\max(\tau,\rho\gamma)}^{(m)}$, $\Delta^{(m)}(b_{i,n,t}) > \rho\gamma > \rho\beta_t^{1/2}\sigma_{t-1}^{(m)}(b_{i,n,t})$. In the fourth step we have used the following facts, $t > u \geq \max\{3, (2(\rho - \rho_0)\eta)^{-2/3}\}$, $M\pi^2/2\delta > 1$ and $\sigma_{t-1}^{(m)}(b_{i,n,t}) > \eta/\sqrt{t}$ to conclude,

$$(\rho - \rho_0)\frac{\eta\sqrt{4\log(t)}}{\sqrt{t}} > \frac{1}{t^2} \implies (\rho - \rho_0) \cdot \sqrt{2\log\left(\frac{M\pi^2 t^2}{2\delta}\right)} \cdot \frac{\eta}{\sqrt{t}} > \frac{1}{t^2}$$

$$\implies (\rho - \rho_0)\beta_t^{1/2}\sigma_{t-1}^{(m)}(b_{i,n,t}) > \frac{1}{t^2}.$$

In the seventh step of (13) we have bound the sum by an integral and used $\rho_0 \geq 2$ twice. Finally, the last step follows by $\rho_0 \geq 1 + \sqrt{(1 + 2/\alpha)/(1 + d)}$ and noting $M \geq 1$. ∎

## D Addendum to Experiments

### D.1 Other Baselines

For MF-NAIVE we limited the number of first fidelity evalutions to $\max\left(\frac{1}{2}\frac{\Lambda}{\lambda^{(1)}}, 500\right)$ where $\Lambda$ was the total budget used in the experiment. The 500 limit was set to avoid unnecessary computation – for all of these problems, 500 queries are not required to find the maximum. While there are other methods for multi-fidelity optimisation (discussed under Related Work) none of them had made their code available nor were their methods straightforward to implement - this includes MF-SKO.

In addition to the baselines presented in the figures, we also compared our method to the following methods. The first two are single fidelity and the last two are mutlti-fidelity methods.

- The probability of improvement (PI) criterion for BO. We found that in general either GP-UCB or EI performed better.
- Querying uniformly at random at the highest fidelity and taking the maximum. On all problems this performed worse than other methods.
- A variant of MF-NAIVE where instead of GP-UCB we queried at the first fidelity uniformly at random. On some problems this did better than querying with GP-UCB, probably since unlike GP-UCB it wasn't stuck at the maximum of $f^{(1)}$. However, generally it performed worse.

- The multi-fidelity method from Forrester et al. [9] also based on GPs. We found that this method didn't perform as desired: in particular, it barely queried beyond the first fidelity.

A straightforward way to incorporate lower fidelity information to GP-UCB and EI is to query at lower fidelities and use them in learning the kernel $\kappa$ by jointly maximising the marginal likelihood. While the idea seems natural, we got mixed results in practice. On some problems this improved the performance of all GP methods (including MF-GP-UCB), but on others all performed poorly. One explanation is that while lower fidelities approximate function values, they are not always best described by the same kernel. The results presented do not use lower fidelities to learn $\kappa$ as it was more robust. For MF-GP-UCB, each $\kappa^{(m)}$ was learned independently using only the queries at fidelity $m$.

## D.2 Description of Synthetic Experiments

The following are the descriptions of the synthetic functions used. The first three functions and their approximations were taken from [32].

**Currin exponential function:** The domain is $\mathcal{X} = [0,1]^2$. The second and first fidelity functions are,

$$f^{(2)}(x) = \left(1 - \exp\left(\frac{-1}{2x_2}\right)\right)\left(\frac{2300x_1^3 + 1900x_1^2 + 2092x_1 + 60}{100x_1^3 + 500x_1^2 + 4x_1 + 20}\right),$$

$$f^{(1)}(x) = \frac{1}{4}f^{(2)}(x_1 + 0.05, x_2 + 0.05) + \frac{1}{4}f^{(2)}(x_1 + 0.05, \max(0, x_2 - 0.05)) +$$

$$\frac{1}{4}f^{(2)}(x_1 - 0.05, x_2 + 0.05) + \frac{1}{4}f^{(2)}(x_1 - 0.05, \max(0, x_2 - 0.05)).$$

**Park function:** The domain is $\mathcal{X} = [0,1]^4$. The second and first fidelity functions are,

$$f^{(2)}(x) = \frac{x_1}{2}\left(\sqrt{1 + (x_2 + x_3^2)\frac{x_4}{x_1^2}} - 1\right) + (x_1 + 3x_4)\exp(1 + \sin(x_3)),$$

$$f^{(1)}(x) = \left(1 + \frac{\sin(x_1)}{10}\right)f^{(2)}(x) - 2x_1^2 + x_2^2 + x_3^2 + 0.5.$$

**Borehole function:** The second and first fidelity functions are,

$$f^{(2)}(x) = \frac{2\pi x_3(x_4 - x_6)}{\log(x_2/x_1)\left(1 + \frac{2x_7 x_3}{\log(x_2/x_1)x_1^2 x_8} + \frac{x_3}{x_5}\right)}, \quad f^{(1)}(x) = \frac{5x_3(x_4 - x_6)}{\log(x_2/x_1)\left(1.5 + \frac{2x_7 x_3}{\log(x_2/x_1)x_1^2 x_8} + \frac{x_3}{x_5}\right)}.$$

The domain of the function is $[0.05, 0.15; 100, 50K; 63.07K, 115.6K; 990, 1110; 63.1, 116; 700, 820; 1120, 1680; 9855, 12045]$ but we first linear transform the variables to lie in $[0,1]^8$.

**Hartmann-3D function:** The $M^{\text{th}}$ fidelity function is $f^{(M)}(x) = \sum_{i=1}^{4} \alpha_i \exp\left(-\sum_{j=1}^{3} A_{ij}(x_j - P_{ij})^2\right)$ where $A, P \in \mathbb{R}^{4 \times 3}$ are fixed matrices given below and $\alpha = [1.0, 1.2, 3.0, 3.2]$. For the lower fidelities we use the same form except change $\alpha$ to $\alpha^{(m)} = \alpha + (M - m)\delta$ where $\delta = [0.01, -0.01, -0.1, 0.1]$ and $M = 3$. The domain is $\mathcal{X} = [0,1]^3$.

$$A = \begin{bmatrix} 3 & 10 & 30 \\ 0.1 & 10 & 35 \\ 3 & 10 & 30 \\ 0.1 & 10 & 35 \end{bmatrix}, \quad P = 10^{-4} \times \begin{bmatrix} 3689 & 1170 & 2673 \\ 4699 & 4387 & 7470 \\ 1091 & 8732 & 5547 \\ 381 & 5743 & 8828 \end{bmatrix}$$

**Hartmann-6D function:** The 6-D Hartmann takes the same form as above except $A, P \in \mathbb{R}^{4 \times 6}$ are as given below. We use the same modification to obtain the lower fidelities using $M = 4$.

$$A = \begin{bmatrix} 10 & 3 & 17 & 3.5 & 1.7 & 8 \\ 0.05 & 10 & 17 & 0.1 & 8 & 14 \\ 3 & 3.5 & 1.7 & 10 & 17 & 8 \\ 17 & 8 & 0.05 & 10 & 0.1 & 14 \end{bmatrix}, \quad P = 10^{-4} \times \begin{bmatrix} 1312 & 1696 & 5569 & 124 & 8283 & 5886 \\ 2329 & 4135 & 8307 & 3736 & 1004 & 9991 \\ 2348 & 1451 & 3522 & 2883 & 3047 & 6650 \\ 4047 & 8828 & 8732 & 5743 & 1091 & 381 \end{bmatrix}$$

Figure 8: The simple regret $S(\Lambda)$ against the spent capitcal $\Lambda$ on the synthetic functions. The title states the function, its dimensionality, the number of fidelities and the costs we used for each fidelity in the experiment. All curves barring DiRect (which is a deterministic), were produced by averaging over 20 experiments. The error bars indicate one standard error.

(a)                                        (b)

Figure 9: (a) illustrates the functions used in the Bad Currin Exponential experiment where we took $f^{(1)} = -f^{(2)}$ and (b) shows the simple regret for this experiment. See caption under Fig. 8 for more details.

## D.3   More Results on Synthetic Experiments

Figure 8 shows the simple regret $S(\Lambda)$ for the synthetic functions not presented in the main text.

It is natural to ask how MF-GP-UCB performs with bad approximations at lower fidelities. We found that our implementation with the heuristics suggested in Section 5 to be quite robust. We demonstrate this using the Currin exponential function, but using the negative of $f^{(2)}$ as the first fidelity approximation, i.e. $f^{(1)}(x) = -f^{(2)}(x)$. Figure 9 illustrates $f^{(1)}, f^{(2)}$ and gives the simple regret $S(\Lambda)$. Understandably, it loses to the single fidelity methods since the first fidelity queries are wasted and it spends some time at the second fidelity recovering from the bad approximation. However, it eventually is able to achieve low regret.

Finally, we present results on the cumulative regret for the synthetic functions in Figure 10.

## Acknowledgements

We wish to thank Bharath Sriperumbudur for the helpful email discussions. This research is partly funded by DOE grant DESC0011114.

Figure 10: The cumulative regret $R(\Lambda)$ against the spent capitcal $\Lambda$ on the synthetic functions. The title states the function, its dimensionality, the number of fidelities and the costs we used for each fidelity in the experiment. All curves barring DiRect (which is a deterministic), were produced by averaging over 20 experiments. The error bars indicate one standard error.