[Reviews · NeurIPS 2016]

Reviewer 1

Summary

This paper presents a novel algorithm to extend the standard GP-based BO to deal with multi-fidelity setting. It provides extensive results both theoretical and experimental.

Qualitative Assessment

Bayesian optimization is a hot topic right now in the NIPS community, and multi-fidelity setups are quite important in applications. I believe it has been mentioned in many workshops and panels. This paper provides an extensive analysis of the topic in the theoretical and experimental side. However, this paper has two important issues from my point of view. First, it is a journal paper condensed in NIPS space. On one hand, this is impressive, because the amount of work presented is quite large. On the other hand, it makes it hard to understand (virtually impossible without the Appendix). The authors expend a large amount of text to give examples (e.g.: the gold robot) and provide intuitive explanations. While this is something that is quite valuable, in this case it makes it hard to understand the actual algorithm and theoretical contribution. Furthermore, this is aggravated by the use of non-conventional notation. For example, in the literature, typically vectors and matrices are bold, scalars are lowercase and sets are capital. This notations is partly followed in the GP description, but not elsewhere. Second, the paper makes strong assumptions with the purpose of finding a theoretical bound, most of them pointed by the authors themselves. Later, in the experiments, the assumptions are found impractical and the algorithm is modified with heuristics. Concerning the assumptions, it is not clear if the "no reward" for lower fidelities is a requirement to simplify the theoretical analysis of it is is actually a requirement of the algorithm to be efficient. A minor assumption that is barely mentioned is that the analysis and the algorithm assumes that the ordering is known. While this is a very reasonable assumption in practice, there might be situations where it is unavailable. My major concern with the algorithm and the analysis is that the idea that every fidelity has to be modeled independently. My intuition about multi-fidelity setups in practice is sampling noisy versions of the actual function. The location of the optimum might be slightly different, but the correlation between fidelities should be high. Thus, why not sharing information between fidelities? Ideally, a single surrogate model should be enough, but at least, some correlation should appear like in multi-tasking or contextual bandits. For example, in Figure 2, the blue model could be easily improved given all the information available. Instead the authors focus on the acquisition function. Finally, the experimental section is impressive. There is only a couple of minor concerns. Clearly, there is a connection between the fidelity "error" and the cost. For the Hartmann function, I could not find how it was tweaked to get the fidelities. Second, the costs of the fidelities seems to be too large (x10 per fidelity). It would be interesting to see how the methods perform with different costs (for example, x2), to find out where it is interesting to start using MF vs single fidelity methods. This last point is a mere suggestion, maybe not even for NIPS due to the lack of space. It was hard to find the cost in the figures. I would explain it in the main text or, at least, use \lambda in the figures as in the rest of the paper.

Confidence in this Review

3-Expert (read the paper in detail, know the area, quite certain of my opinion)


Reviewer 2

Summary

A relevant and well-written paper, which proposes an extension to the Bayesian Optimization framework in which multiple surrogate/acquisition functions - each with its own fidelity - are integrated in an extended UCB procedure. The paper provides a thorough theoretical analysis of the procedure resulting a promising bound under a specific formulation of the regret. Further, the paper evaluates the algorithm on a number of synthetic examples a three examples hyper parameter optimization. The derived bound and the evaluation show the efficiency of the approach compared to traditional GP-UCB and a number of other alternatives (under the specific regret).

Qualitative Assessment

Technical quality: - The paper provides a very thorough and convincing analysis together with a number of numerical examples to guide the reader. - I suppose an alternative to combining the various functions in the decision stage, would be to explicitly model the fidelity/confidence of a particular observation via a special (composite) likelihood, possibly with multiple GP priors…? Novelty/originality - The paper explores a relatively new idea; therefore, I would have expected a bit more emphasis on existing works within the paper itself to clarify the actual novelty. However, given the amount of work to describe, I accept the current compromise of placing the comparison in the appendix. - While the idea of multifidelity has been explored in previous work the current work provides an important step forward in providing a formal regret and in particular derives a regret bound for the specific procedure. - I am sure the procedure will be picked up by practitioners in the field. Clarity and presentation - The paper is quite dense and technical, yet I find that it has a suitable mix of novel models, theoretical/numerical analysis and experimental results to make it of general interest to the NISP community. - My only concern is the extensive use of references to the appendix which disturbs the flow of the paper.

Confidence in this Review

2-Confident (read it all; understood it all reasonably well)


Reviewer 3

Summary

The authors considered Bayesian optimization framework. The main difference of the considered problem statement consists in that multi-fidelity black-boxes are available, realizing the same function. The data source with the highest fidelity at the same time is the most expensive to evaluate. It is assumed that costs of evaluation of data sources with different fidelities are given. The authors proposed multi-fidelity bandit problem statement, they formalized the notion of regret when explicit costs of different fidelity observations are given. Then the authors proposed an approach for Bayesian optimization, which explicitly takes into account costs of observations and relationships between black-boxes with different fidelities. The approach is based on Gaussian process upper confidence bound algorithm. Asymptotics of the corresponding regret is provided w.r.t. increasing capital of a resource, available for black-boxes evaluations. Extensive computational experiments provide evidence about efficiency of the proposed approach.

Qualitative Assessment

Comments: - line 88. It seems that the comma should be placed before the word «where», not after this word. - line 101. In the formula for a Gaussian distribution (at the end of the line) there is a redundant power «2». It seems that after the word «with» the comma should not be used. - line 105. The definition of the regret depends on B, which is defined in A3. In practice, if I would like to monitor the dependence of the regret on iterations number, how should I set a value of B? - In formula (2) the authors introduce variables $\lambda^{(m)}$. However, starting from line 119 there is no any explanation what $\lambda^{(m)}$ stands for (although it is quite easy to understand, but formally it should be written). - Will the considered regret bound and theoretical results about it be still valid if for the case of lower fidelity functions we set the instantaneous rewards to be equal not to -B, but to some set of arbitrary numbers? - Formula (3). At the end of the formula the dot should be placed. - line 252. «...if the algorithm, is stuck at fidelity m for too long then …». What is «too long» here? Any possibility to make a decision automatically? Or a human intervention is needed? - Are there any recommendations how to set values of $\lambda^{(m)}$? Of course, I understand, that these values are defined by to which extent a high fidelity model is more expensive to evaluate then low fidelity models. However, this information is not always precisely known. At the same time values of $\lambda^{(m)}$ influence the outcomes of the proposed algorithm. That is why I am asking about the corresponding authors' experience. - line 313, Type Ia Supernovae. M = 3 fidelity functions are considered. In previous examples these functions were defined e.g. by the sample size, used to estimate the functions values. How is it done in this test case? - line after line 396 (Appendix, A.1 Table of notations), the seventh row from below. There is some problem with a reference to some equation. - line 491. «Let $\sigma_{t-1}$ denote the posterior…» -> «Let $\sigma_{t-1}^2$ denote the posterior..." - line 532. «vol(vol(…»-> «vol(…» - In the proof of theorem 10 I would like to ask the authors to explicitly: a) write where in the proof assumption A2 (line 105) is used, b) comment on whether we consider the statement of the theorem conditional only on those GP trajectories, which fulfill assumptions A2 and A3 (line 105), or not. Conclusions: - The topic of the paper is very important. E.g. in engineering practice it is a typical situation when black-boxes of different fidelities are available. - As far as I know this is the first time when a notion of a cumulative regret for the multi-fidelity case is rigorously introduced. - The proofs seem correct. Results of experiments confirm theoretical conclusions. The paper is well written. - Therefore, I would recommend to publish this paper in the NIPS proceedings.

Confidence in this Review

3-Expert (read the paper in detail, know the area, quite certain of my opinion)


Reviewer 4

Summary

The paper proposes a model referred to as the multi-fidelity framework for which a decision maker sequentially selects a point together with a fidelity level in the hope of maximizing its cumulative reward. Fidelity approximations correspond to cheap evaluations of the expensive function to be evaluated. At each round, The DM receives a bandit feedback that is used to update Gaussian Process fidelity posteriors. The authors prove an upper bound on their proposed algorithm. The algorithm is compared empirically to various baselines both on synthetic and real-world datasets.

Qualitative Assessment

The paper is very well written and easy to follow. Clear and complete description of the algorithm has been provided. The problem setting is clearly explained and well motivated through various convincing application examples. The model is described in details and the help of figures to support key points is appreciated, making the reading enjoyable. The main contributions of the paper are the formalism for the novel multi-fidelity bandit optimization setting and the MF-GP-UCB algorithm for which a theoretical analysis of the regret upper bound is provided. The algorithm tries to discard bad regions at low fidelities while it successively concentrates on promising regions at higher fidelities. The upper bound provided for the payoff function f is chosen as the minimum of the fidelity upper bounds. I wonder whether one could not aggregate feedback for every fidelity to obtain a tighter upper bound. The authors discuss terms appearing in the obtained upper bound and compare them to the GP-UCB one. The last section is devoted to experiments. These are well motivated and the baselines used to evaluate MF-GP-UCB are convincingly chosen. Typo: - l.11: settting -> setting AFTER REBUTTAL I have read the rebuttal and kept my scores.

Confidence in this Review

2-Confident (read it all; understood it all reasonably well)


Reviewer 5

Summary

The paper provides an algorithm for surrogate based optimization in case of multifidelity data. The main contributions are new regret function for a multifidelity setting, theoretical justification for provided algorithm as an upper bound for regret, and implementation of the proposed algorithm.

Qualitative Assessment

The paper presents novel results and is applicable to a wide range of multifidelity engineering optimization problems. Authors contributes significantly to the field of multifidelity optimization. Approach to define fidelity of function in this paper is different from what is used commonly in the literature: it assumes that we know how much low fidelity function can differ from the highest fidelity function, moreover, we know the bound for the high fidelity function. As authors suggest at l.247 to apply these assumptions in a practical problem we need to simplify them. Note that introduced assumptions also prohibit analytically tractable inference if applied - and we need to do some kind of approximation. In many applications observations are subject to noise - for example, if we do cross-validation we rarely have smooth enough dependence of quality functional on parameter to optimize. The paper considers noise free algorithm. While MIG states that some noise is presented, provided experiments don't adress noise variance selection. Often optimization of GP kernel leads to degenerate solution. Typically, one uses some kind of regularization of kernel parameters. It would be benefitial if authors state what GP library they use for experiments. Amount of information in the paper and supplementary materials is huge, which degrades clarity of representation. For example, authors can limit themself to the case of only two fidelities to simplify description of results. Also due to lack of space authors often uses construction "bold_title: ......" to avoid placing a subsection bold_title at a separate line instead. Techincal improvements suggestion: l.247 - This corresponds to assuming ... - not true. In fact less strong assumption than this holds. l.326 - Conclusion should be a separate section, not a part of Experiments. l.332 - Reference style is not consistent: in some cases there are full name of authors ([1], [2]), sometimes - initials with dots ([3], [22]), sometimes - initials without dots ([9]). Journal papers lack volume and issue information. Journal names are also not consistent: sometimes they are shortened for brevity (J.Mach.Learn.Res., l.334), sometimes not (Structural and Multidisciplinary optimization) To summarize: presented results are sound and important, while the current version of paper is more like a journal-type paper, and doesn't fit into the conference format.

Confidence in this Review

2-Confident (read it all; understood it all reasonably well)


Reviewer 6

Summary

This paper consider the problem of maximizing an expensive noisy function in the bandit setting. Furthermore, multiple cheap approximations (called fidelities) of the expensive function are available. By assuming a gaussian process prior, the paper provides an algorithm called MF-GP-UCB that uses extensively the cheap approximations in order to evaluate as low as possible the expensive function before convergence. Theoretical analysis of the regret of MF-GP-UCB are provided together with synthetic and real experiments. The proposed algorithm is compared against some other baselines where one can see that MF-GP_UCB used less time to achieve a desired accuracy.

Qualitative Assessment

1- Isn't assumption A1 too strong and unrealistic? I mean by assuming that all fidelities are sampled from the same gaussian process it makes the problem somehow easier and not so interesting. For example if the true function is sin(x) and the other fidelities are the taylor approximations , then would the assumption holds in this case? 2- I have the impression that the approximative fidelities function used for Currin exponential and other are not so computationally less intensive. It seems even more computationally intensive by calling multiple instance of the original function. This is in contraction with the motivations of the paper. 3- Provide a clear mathematical description of the function to optimise and the approximative fidelities for the real experiments 4- Mention in the paper that the exact form of the function to optimise for the synthetic experiments is provided in the appendix.

Confidence in this Review

1-Less confident (might not have understood significant parts)